# Heat wave characteristics: evaluation of regional climate model performances for Germany

Dragan Petrovic[1], Benjamin Fersch[1] and Harald Kunstmann[1,2]

[1]Institute of Meteorology and Climate Research (IMK-IFU), Karlsruhe Institute of Technology, Campus Alpin, Kreuzeckbahnstraße 19, 82467 Garmisch-Partenkirchen, Germany
[2]Institute of Geography and Center for Climate Resilience, University of Augsburg, Alter Postweg 118, 86159 Augsburg, Germany

*Correspondence to*: Dragan Petrovic (dragan.petrovic@kit.edu)

**Abstract.** Heat waves are among the most severe climate extreme events. In this study, we address the impact of increased model resolution and tailored model settings on the reproduction of these events by evaluating different regional climate model outputs for Germany and the near surroundings between 1980–2009. Both, outputs of an ensemble of six EURO-CORDEX models of 12.5 km grid resolution and outputs from a high resolution (5 km) WRF run are employed. The latter was especially tailored for the study region regarding the physics configuration. We analyze the reproduction of maximum temperature, number of heat wave days, heat wave characteristics (frequency, duration and intensity), the 2003 major event and trends in the annual number of heat waves. E-OBS is used as reference and we imply Taylor diagram, Mann-Kendall trend test, spatial efficiency metric and cumulative heat index as a measure for intensity. Averaged over the domain, heat waves occurred about 31 times in the study period with an average duration of 4 days and average heat excess of 10 °C. Maximum temperature was reproduced reasonably well by all models. Despite the same forcing, the models exhibited a large spread in the heat wave reproduction. The domain mean conditions of heat wave frequency and duration were captured reasonably well, but intensity was reproduced weakly. The spread was particularly pronounced for the 2003 event, indicating the difficulty of models to reproduce single major events. All models underestimated the spatial extent of the observed increasing trends. WRF mostly did not perform significantly better than the other models. We conclude that increased model resolution does not add a significant value to heat wave simulation if the base resolution is already relatively high. Tailored model settings seem to play a minor role. The partly distinct differences in performance, however, highlight that the choice of model can be crucial.

## 1 Introduction

Heat waves are climatological extreme events with severe negative impacts on organisms and ecosystems. Their effects can be illness, large-scale mortality, substantial losses in agricultural production, forest fires and increased energy demand for cooling (Beniston et al., 2007; De Bono et al. 2004; Ciais et al., 2005; Robine et al., 2008; Kyselý et al., 2011; Bastos et al., 2014; Urban et al., 2017). Especially in the midlatitude zones, heat waves are regarded as a major cause of weather-related

human mortalities (Luber, 2008; Plavcová and Kyselý, 2019). In Europe, they are a regular part of the summer climate (Vautard et al., 2013), and an increasing trend in the number of summer heat waves was reported for the recent decades with prolonging tendency (Della-Marta et al., 2007; Kyselý, 2010; Valeriánová et al., 2015; Saeed et al., 2017). According to Fischer and Schär (2010), the frequency and length of heat waves have nearly tripled and doubled, respectively, over the period 1880 – 2005 in Western European regions. Especially since the millennium, Europe experienced multiple extraordinary heat waves (Lhotka et al., 2018b). Two of the most prominent events were the heat episodes in 2003 over Western Europe (Fink et al., 2004) and the 2010 event over Eastern Europe and Russia (Schneidereit et al., 2012). In the more recent past, severe summer heat episodes took place in Central Europe in 2013 (Lhotka and Kyselý, 2015b), 2015 (Hoy et al., 2016) and especially in 2018 (e.g., Vogel et al., 2019, Rousi et al., 2022). In the context of climate change, heat waves are expected to become more frequent, more intense and longer lasting in the future around the globe (Meehl and Tebaldi 2004; Lau and Nath 2014; Lemonsu et al. 2014; Seneviratne et al., 2014; Diffenbaugh and Ashfaq 2010). Coumou and Rahmstorf (2012) estimate the probability of severe events like 2003 over Western Europe has increased by a factor of 2 – 4 because of global warming. Shifts of the temperature distribution are considered as the primary drivers of these changes (Ballester et al., 2010; Lau and Nath, 2014). According to Fischer and Schär (2010), larger variance of summer temperature distribution in future climates are also possible drivers. Lhotka et al. (2017) additionally emphasize increased temporal autocorrelation of daily maximum temperature, which leads to more persistent heat waves, as potential drivers.

To analyze and understand such changes at the regional scale and to allow projections for future characteristics of the heat wave events, regional climate models (RCMs) are used. For better interpretation of future climate scenarios and their uncertainties and limitations, it is important to evaluate RCM simulations for the recent or historical climate to detect biases, which are usually present, despite the added value compared to GCMs (Lhotka et al., 2018a; Plavcová and Kyselý, 2019; Lin et al., 2022). The availability and reliability of RCM simulations have rapidly evolved in the last years (Štepánek et al., 2016). This is also due to concerted downscaling projects and initiatives such as PRUDENCE (Christensen and Christensen, 2007), ENSEMBLES (van der Linden and Mitchell, 2009) and most recently CORDEX (Giorgi et al., 2009). In the recent past there have been several heat wave related studies using data from the CORDEX initiative for different parts of the world with focus on both, past periods and future scenarios. For the EURO-CORDEX domain, a focus was put on the evaluation of the RCM's capability in simulating past heat episodes in France (Ouzeau et al., 2016) and all of Europe (Vautard et al., 2013; Lhotka et al., 2018a; Plavcová and Kyselý, 2019; Lin et al., 2022) as well as on the development of future episodes under different scenarios for Portugal (Cardoso et al., 2019), France (Ouzeau et al., 2016), the Mediterranean area (Molina et al., 2020) and all of Europe (Lhotka et al., 2018b; Smid et al., 2019; Machard et al., 2020; Lin et al., 2022). Regarding the rest of the globe and the other CORDEX domains, evaluation studies have been carried out for Africa (Russo et al., 2016), East Asia (Wang et al., 2019b) and South America (Silva et al., 2022). Furthermore, projection studies were performed for Africa (Dosio, 2017), Afghanistan (Aich et al., 2017), South America (Feron et al., 2019), China (Wang et al., 2019a), the MENA region (Varela et al., 2020), the Eastern Mediterranean (Wedler et al., 2023), East Asia (Kim et al., 2023) as well as for the entire globe (Coppola et al., 2021). In the mentioned studies, different horizontal grid resolutions of the models were used and the effects of increased

resolution were often analyzed, which led to different findings: Zeng et al. (2016) and Vichot-Llano et al. (2021), for example, found that higher resolution leads to better reproduction of temperature fields, while Di Luca et al. (2013) came to the conclusion that the potential for added value of increased resolution is small. It is important to consider the differences between the compared resolutions in such studies (Petrovic et al., 2022). Besides the model resolution, the model setup regarding the domain configuration and physical parameterizations for the selected target region is a crucial factor for reliable simulations (e.g., Stoelinga et al., 2003; Kumar et al., 2010). For the temperature simulation, Vautard et al. (2013) found that it is primarily sensitive to convection and microphysics schemes. They emphasize that a large part of the model spread in their study can be attributed to different parameterizations. Moreover, they draw a connection between parameterizations and different spatial resolutions. Mooney et al. (2013) found that simulated temperature showed relatively high sensitivity to the land surface models, some sensitivity to the radiation schemes and little sensitivity to the microphysics and planetary boundary layer (PBL) schemes. They concluded a strong dependence on the region and season of the optimal parameterization combination. Kotlarski et al. (2014) state that bias spreads between different configurations of the same model can be similar to those between different models.

To our knowledge, there is no study that presents an evaluation of the EURO-CORDEX RCM's capability to reproduce heat wave characteristics for Germany, which has motivated us to perform this analysis. At the same time this study is the follow-up to the model comparison study for droughts from Petrovic et al. (2022). The thematic proximity is obvious, since heat and drought are often, but not always, related to each other. We analyze a variety of RCM simulations, i.e., a 5 km three-domain Weather Research and Forecasting Model (WRF, Skamarock et al., 2008) run and an ensemble of six EURO-CORDEX realizations at 12.5 km horizontal resolution. The setup for the WRF model was thoroughly determined for Germany. Intuitively, one would expect better performance in simulating hot temperature from WRF due to the higher resolution and the focus on the target region compared to the EURO-CORDEX runs. The WRF model was shown to be capable of simulating spatiotemporal features of heat wave events over a large domain (Wang et al., 2019b). To attribute to a resolution or settings effect of the WRF model performance, we additionally include a 15 km simulation configuration in our analysis which is slightly coarser than for the EURO-CORDEX standard. Therefore, following up on the drought study (Petrovic et al., 2022), the objectives of this study are:

1. Evaluation of the performance of regional climate models in reproducing hot temperatures and associated heat wave characteristics by employing a six-member EURO-CORDEX ensemble and a high resolution WRF run. The EURO-CORDEX RCMs and WRF differ in resolution, while the model physics configurations differ among every single RCM.

2. Obtaining insights into the heat wave course for Germany and its near surrounding between 1980 and 2009.

For this purpose, the results are evaluated and compared to observations. Specifically, we analyze reproduction of daily maximum temperature, the number of heat wave days, heat wave characteristics (frequency, duration and intensity), trends in the number of heat waves per year and the heat wave event in 2003.

Moreover, we can compare the new core findings for heat waves with those from the aforementioned drought analysis (Petrovic et al., 2022) in terms of similarities and differences.

## 2 Data

In this study, data from the same sources as in Petrovic et al. (2022) is used. While in that study monthly data of precipitation, minimum and maximum temperature was used, here the daily values of the maximum (surface) temperature ($T_{max}$) are employed.

We use data from an ensemble of six EURO-CORDEX RCM simulations. Each of the experiments were conducted with 0.11° ($\approx$ 12.5 km) horizontal grid resolution and cover the EUR-11 CORDEX-Domain. Outputs from the following RCMs were used: COSMO-CLM, ALADIN 6.3 (hereafter referred to as ALADIN in the text), REMO2015 (REMO), RegCM 4.6 (RegCM), RACMO 2.2e (RACMO) and RCA4. All the runs obtained the boundary conditions from the global ERA-Interim reanalysis (Dee et al, 2011). When this analysis was initiated, these runs were the only ones that cover the study period 1980–2009 and contain the relevant meteorological variables.

In addition to the EURO-CORDEX data, we included the outputs from the ERA-Interim forced reanalysis WRF run for the time period 1980 – 2009 from Warscher et al. (2019). These simulations were preceded by a comprehensive search and final identification of optimal model physics and parameterization configuration for the target region Germany (Wagner and Kunstmann, 2016). This is the first major difference to the EURO-CORDEX outputs, which were aimed at the entire EUR-11 CORDEX-domain (Giorgi et al., 2009). A two domain setup with one-way nesting was employed to downscale the ERA-Interim reanalysis of approx. 75 km. The horizontal grid resolution of the innermost domain, which frames Germany and the near surroundings, is 5 km. This increased resolution is the second major difference to the EURO-CORDEX outputs. As mentioned above, we also used the outputs from the 15 km second WRF domain of the same run. Therefore, we will refer to WRF@5 km and WRF@15 km from here on to distinguish between the two runs.

More detailed information about the EURO-CORDEX RCMs and an overview of the different model physics configurations for all runs can be obtained from Table 1 and 2 in Petrovic et al. (2022). The gridded observational data set from E-OBS (Haylock et al., 2008), version 23.1e, in 0.1° ($\approx$ 11.1 km) horizontal grid resolution serves as reference. The study region extends from 6° to 15° E and 47° to 55° N, so that it contains Germany and its near surroundings. The WRF and E-OBS data were regridded using bilinear interpolation to match the horizontal grid resolution of the EURO-CORDEX RCMs.

## 3 Methods

### 3.1 Analysis of daily maximum temperature reproduction

Since $T_{max}$ is the main variable determining heat waves, the grid cell based summer values (June, July, August) are first analyzed. We use the Taylor diagram (Taylor, 2001), which provides a succinct visual statistical summary in terms of agreement between patterns regarding their correlation, their root-mean-square difference, and the ratio of their variances or standard deviations (Taylor, 2001). Moreover, we calculate the density plot to visualize and compare the distributions of the

values of the individual data sets. From spatially and temporally averaged daily values, we calculate the mean bias values to
be able to draw conclusions about the role of the model bias for the further results. It must be noted that not only a good
simulation of the right tail of the temperature frequency distribution is of importance for the reproduction of heat waves, but
also the persistence of the high temperatures (Lhotka et al., 2018a).

## 3.2 Heat wave definition

There is no universal definition of heat waves. In fact, there are multiple definitions that include different metrics and criteria
depending on the region, season and purpose of the study (Feron et al., 2019; Becker et al., 2022). Generally, it describes a
period of consecutive days with conditions excessively higher than normal (Perkins et al., 2012). Here, we define a heat wave
as an event of at least three consecutive days where the 90[th] percentile of $T_{max}$ based on each calendar day of the study period
is exceeded (Fischer and Schär, 2010). Therefore, the 90[th] percentile for each calendar day and each grid cell from each data
set was calculated first. This was done for each dataset individually to circumvent the $T_{max}$ biases among the different models
(Vautard et al., 2013; Lhotka et al., 2018b). We only addressed summer heat waves in this study.

## 3.3 Analysis of heat wave characteristics

Based on the heat wave definition described above, we calculate the number of heat wave days and number of heat waves for
each grid cell for the whole study period 1980 – 2009. Based on the number of heat wave days and heat waves, we determine
the mean duration of heat waves for each grid cell from each data set. In order to describe the mean heat wave intensity, we
use the cumulative heat index as a measure (e.g. Katavoutas and Founda, 2019; Perkins-Kirkpatrick and Lewis, 2020; Founda
et al., 2022). It refers to the integration of heat exceedance over the 90[th] percentile threshold for all heat wave days during a
heat episode or whole season:

$$CumHeat = \sum_{i=1}^{n} \Delta(T_{max,i} - T_{max,P90,i}) \qquad (1)$$

where $i$ indicates the calendar day of the heat wave event and $T_{max,P90,i}$ the 90[th] $T_{max}$ percentile of day $i$, for each grid cell. To
get the mean intensity of heat waves, we integrated all excess values for the whole study period and divided the results by the
number of heat waves for each grid cell of each data set. For each of the single aspects (number of heat wave days and heat
waves, mean heat wave duration and mean heat wave intensity) we calculate the domain mean value for every data set. In the
next step, we subtract the E-OBS reference domain from each RCM domain to get the bias patterns and also calculated the
domain mean values. To further evaluate the spatial agreement between reference and each RCM, we utilize the spatial
efficiency (SPAEF) metric (Demirel et al., 2018; Koch et al., 2018). The SPAEF is a multiple components performance metric
for the comparison of spatial patterns. It is calculated as:

$$SPAEF = 1 - \sqrt{(\alpha - 1)^2 + (\beta - 1)^2 + (\gamma - 1)^2} \qquad (2)$$

with the three components α as the Pearson correlation coefficient between observed (obs) and simulated (sim) patterns,

$$\beta = (\frac{\sigma_{sim}}{\mu_{sim}})/(\frac{\sigma_{obs}}{\mu_{obs}}) \qquad (3)$$

as the fraction of coefficient of variation representing spatial variability and

$$\gamma = \frac{\sum_{i=1}^{n} \min(K_i, L_i)}{\sum_{i=1}^{n} K_i} \tag{4}$$

as the overlap between the histograms of the observed ($K$) and simulated patterns ($L$), both containing the same number $n$ of bins. For the calculation of $\gamma$ the z score of the patterns is used, which allows comparison of two variables with different units. For both histograms of $K$ and $L$, the number of values in each bin $i$ is counted. Afterwards for each bin the lower (minimum)

number of $K_i$ or $L_i$ is picked, which indicates the number of shared values in the same bin. Thereafter these numbers are summed up and divided by the total number $n$ of values in $K$ or $L$. The SPAEF has a predefined range between -∞ and 1, where 1 corresponds to ideal agreement between two patterns. The three components are independent of each other and typically equally weighted so that they complement each other in a meaningful way and provide holistic pattern information. By this way, instead of exact values on the grid scale, global features such as distribution and variability are evaluated (Koch et al.,

175    2018).

### 3.4 Heat wave trend analysis

In order to investigate the temporal characteristics of heat waves occurrences, we apply the non-parametric Mann-Kendall trend test approach (Mann, 1945; Kendall, 1975). For this purpose, we first count the number of heat waves per year for each grid cell to obtain the annual development. Then on the resulting time series of each grid cell the test is applied to detect

significant monotonic trends at a significance level of 0.05. The Mann-Kendall test is based on the correlation between the ranks of a time series and their time order.

For a time series $x_1, x_2, x_3 \dots x_n$, the Mann-Kendall test statistic $S$ is given by

$$S = \sum_{i=1}^{n-1} \sum_{j=i+1}^{n} sign(x_j - x_i) \tag{5}$$

with

$sign(x_j - x_i) = sign(R_j - R_i) = 1$ if $x_j - x_i > 0$     (6)

$sign(x_j - x_i) = sign(R_j - R_i) = 0$ if $x_j - x_i = 0$     (7)

$sign(x_j - x_i) = sign(R_j - R_i) = -1$ if $x_j - x_i < 0$     (8)

where *sign* represents an indicator function, $n$ the number of data points and $R_i$ and $R_j$ their respective ranks. A positive $S$-statistic indicates an increasing trend, a negative one indicates a decreasing trend.

### 3.5 The 2003 heat wave event

The heat wave and drought event in the summer months of 2003 in Central Europe is considered as one of the most severe extreme events in the last decades. It has caused 70,000 excess deaths (Poumadere et al. 2005; Robine et al., 2008), distinct decrease of plant productivity and crop failures (Bastos et al., 2014) and record breaking loss of Alpine glaciers' mass (De Bono et al., 2004). This is why the event is also considered a 'mega heat wave' (Barriopedro et al. 2011; Vautard et al., 2013).

We investigate the capability of the RCMs to reproduce such a single extreme event in terms of intensity and maximum duration. Therefore, we calculate the cumulative heat for the whole summer period 2003 and determined the maximum duration of the heat episodes during this time for each grid cell. The results in this section are also evaluated based on the domain mean values, mean bias values and the SPAEF.

## 4. Results

### 4.1 Daily maximum temperature reproduction

Figure 1 shows the Taylor diagram of the grid cell based $T_{max}$ values from each data set for the summer months. There is an obvious difference between the EURO-CORDEX RCMs and the two WRF runs. In terms of correlation with the E-OBS reference, the two WRF runs stand out with values above 0.9. The 15 km WRF run shows a slightly higher value than its 5 km counterpart. The EURO-CORDEX RCMs reach values between 0.65 and 0.8. RACMO holds the highest value, RegCM the
lowest. Regarding the centered root mean square error (CRMSE), all RCMs have a value below 5. Again, there is a significant difference between the WRF runs and the EURO-CORDEX RCMs, since the WRF runs have distinctly lower values below 2. The 15 km run is slightly better than its 5 km counterpart. The values of the EURO-CORDEX RCMs are relatively close to each other. RACMO holds the lowest CRMSE value, and ALADIN the highest. As far as the agreement of the standard deviation with the reference is concerned, there is a large discrepancy among the EURO-CORDEX RCMs. RACMO comes
closest to the reference, ALADIN's standard deviation shows the biggest difference. RACMO's standard deviation shows even a greater match than the WRF runs, the same goes for COSMO-CLM. Once again, the WRF@15 km run is closer to the reference than its 5 km counterpart. This underlines that the temporal variability is better captured in the coarser resolutions. The results suggest that the WRF model settings are determining for the better performance compared to the EURO-CORDEX RCMs. RACMO is the best performing EURO-CORDEX RCM, ALADIN the worst.

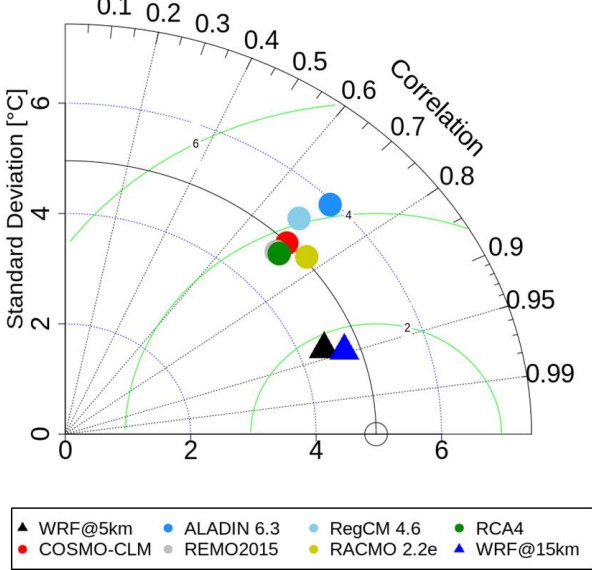

**Figure 1.** Taylor diagram comparing the model performances in reproducing the daily summer $T_{max}$ values in relation to the E-OBS reference for the study period 1980–2009 and the whole study area.

Figure 2 displays the density plot of the summer $T_{max}$ values from each data set. There are pronounced differences between the single distributions in general, but also at the right tail, which is in focus here. Until approx. 10 °C all distributions are relatively similar, the discrepancies begin hereafter. Compared to E-OBS, RCA4 and RACMO are clearly shifted leftwards. Apart from these two models, all other data sets have most of their values in the range between 20 and 25 °C. ALADIN and RegCM are shifted towards the right compared to E-OBS, especially in the right tail area from approx. 30 °C on, in which they clearly have more values than all other runs and E-OBS. In this area, REMO shows high agreement with E-OBS, while WRF@5 km and RCA4 have the least amount of values here. The overall differences between the two WRF runs are distinct. The 15 km run is closer to E-OBS than its 5 km counterpart and it has more values in the right tail of its distribution.

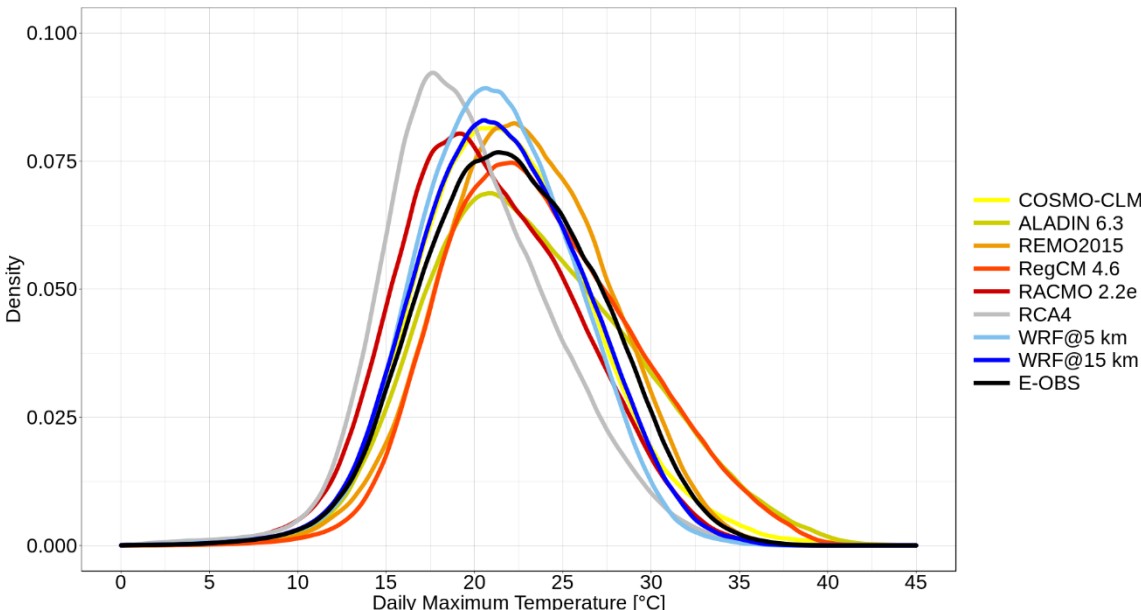

**Figure 2.** Density plot of the summer $T_{max}$ values from each data set.

Table 1 gives the bias values of the spatially and temporally averaged maximum temperature values for each RCM. Five runs, COSMO-CLM, RACMO, RCA4 and the two WRF runs, show negative mean bias values, the other runs positive ones. The
highest negative value is found at RCA4 (-2.40 °C), which is also the highest overall value, followed by RACMO (-1.37 °C). This could be inferred by the density plot in Figure 2. This is also true for the highest positive bias values of ALADIN (1.19 °C) and RegCM (1.62°C). COSMO-CLM (-0.16 °C) holds the lowest bias value. The overall spread is 4.02 °C between RegCM and RCA4.

Comparing the outcomes of the Taylor diagram (Figure 1) with the mean bias values from Table 1 leads to the following
insights: ALADIN (1.19 °C), RegCM (1.62 °C) and RACMO (-1.37 °C) hold relatively large mean bias values, while in the Taylor diagram their scores distinctly differ. Here, ALADIN also has a relatively high CRMSE and low agreement with the reference standard deviation, while RACMO shows the lowest EURO-CORDEX CRMSE, high standard deviation agreement and additionally the highest correlation value from the EURO-CORDEX RCMs. RegCM has also a low correlation value in the Taylor diagram. COSMO-CLM has the lowest mean bias values (-0.16 °C) and additionally shows the highest agreement
with the reference standard deviation in the Taylor diagram. Moreover, it is striking that RCA4 holds the largest mean bias value (-2.40 °C), while it is placed relatively well in the Taylor diagram. For the WRF outputs it is the opposite case. In the Taylor diagram they show highest correlation values as well as lowest CRMSE values, while their mean values in Table 1 are somewhere in the middle. This basically shows that in some case the individual models have strong or weak values in both terms, while in other cases the performances diverge, meaning that the mean bias values are low but the scores in the Taylor
diagram are weak or vice versa. It needs to be considered that for the Taylor diagram all the grid cell $T_{max}$ values were used,

while in Table 1, as mentioned above, the bias values of the spatially and temporally averaged $T_{max}$ values are given. Moreover, it needs to be taken into account that in the CRMSE any mean bias is implicitly corrected.

**Table 1.** Spatially and temporally averaged daily $T_{max}$ bias values with respect to E-OBS.

| Model | $T_{max}$ Bias [°C] |
|---|---|
| COSMO-CLM | -0.16 |
| ALADIN 6.3 | 1.19 |
| REMO2015 | 0.70 |
| RegCM 4.6 | 1.62 |
| RACMO 2.2e | -1.37 |
| RCA4 | -2.40 |
| WRF@5 km | -0.93 |
| WRF@15 km | -0.58 |


## 4.2 Number of heat wave days

Figure 3 presents the E-OBS pattern of the number of heat wave days for the time period 1980 – 2009 along with the grid cell based differences between each RCM and E-OBS. Table 2 provides more detailed information.

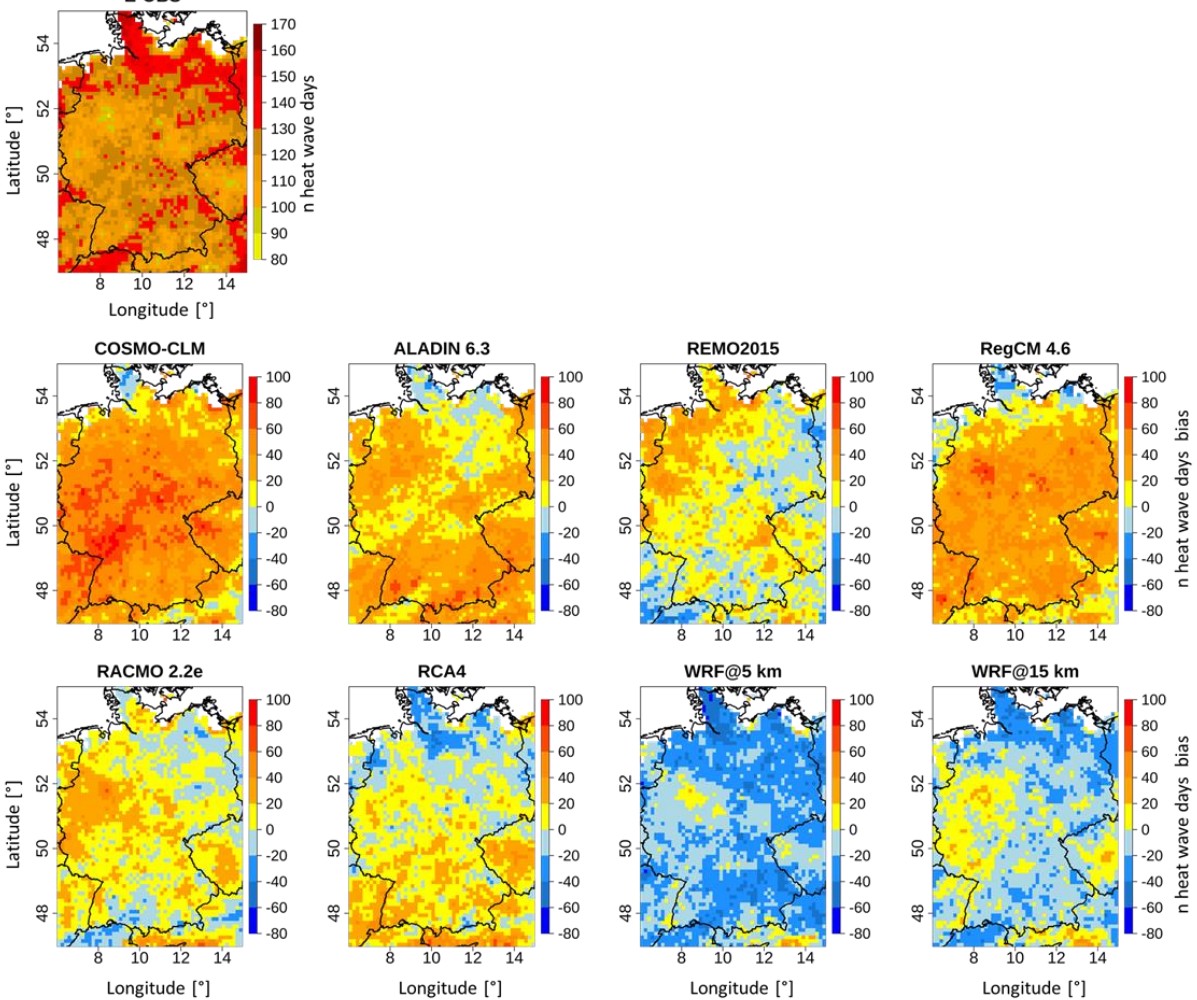

**Figure 3.** Grid cell based E-OBS number of heat wave days pattern for the summer months between 1980–2009 and differences between each RCM and E-OBS.

In the E-OBS domain the highest values are located in the northern, northeastern and southwestern parts with up to 160 heat wave days. The minimum values range between 80 and 90 days and are distributed punctually all over the domain. There is no clear area characterized by low values. The domain mean value is 122 days (Table 2). The RCM difference patterns show distinct differences among each other. It is noticeable that some of the domains either largely show a negative bias (WRF@5 km), which means that they simulated less heat wave days compared to the reference, or a positive bias (COSMO-CLM, ALADIN and RegCM), meaning that they simulated more heat wave days. REMO, RACMO and RCA4 have a rather mixed domain. It is striking that the two WRF outputs are the only ones predominated by negative values. The majority of the values across all domains range between -60 to 60 days of difference. COSMO-CLM, ALADIN, RegCM and RACMO have relatively similar bias values in the western parts of the domain. Other than that, there are no repeating patterns across a majority of the

RCMs. In the WRF@15 km simulation there are significantly more positive bias areas compared to its 5 km counterpart, which is also confirmed in Table 2, where the mean bias value of the 15 km run is much closer to 0 (-8.7 compared to -20.9 days). These positive bias areas are mainly located in the western, eastern and southeastern parts of the domain. The values in Table 2 confirm the impressions from Figure 3: the domain mean values from all EURO-CORDEX RCMs are above the E-OBS

reference value (122 days), with the COSMO-CLM value showing the biggest difference (42 days). The two WRF runs are below the reference (102 and 114 days). The values of REMO and RCA4 (both 130 days) and WRF@15 km come closest to the reference. Regarding the mean bias values, the inferred negative values from the two WRF runs are visible, while the EURO-CORDEX RCMs all show positive mean bias values. COSMO-CLM has the by far the highest bias value (41.8 days), REMO shows the lowest value (7.3 days). It should be kept in mind here that for RCMs that are not dominated by one bias

direction, the values can cancel each other out and provide a small overall mean bias. This is the case for REMO, RACMO and RCA4. The SPAEF values give information about the pattern agreement between the reference and the individual RCMs (not shown here). There is not a single high value. The values are either negative or very low, meaning that there is no good overall spatial agreement from any RCM with the reference. REMO has the highest value (0.19), WRF@ 15 km the lowest (-0.19).


**Table 2.** Number of heat wave days metrics.

| Model | Mean [n days] | Mean Bias [n days] | SPAEF |
|-------|---------------|--------------------|-------|
| COSMO-CLM | 164 | 41.8 | -0.09 |
| ALADIN 6.3 | 149 | 26.2 | 0.03 |
| REMO2015 | 130 | 7.3 | 0.19 |
| RegCM 4.6 | 153 | 31 | -0.15 |
| RACMO 2.2e | 132 | 9.4 | 0.07 |
| RCA4 | 130 | 7.8 | 0.01 |
| WRF@5 km | 102 | -20.9 | -0.11 |
| WRF@15 km | 114 | -8.7 | -0.19 |
| E-OBS | 122 | | |

There are no apparent benefits of the WRF runs visible compared to EURO-CORDEX RCMs. This suggests that neither the increased grid resolution nor the model setup have a decisive effect on the reproduction on the number of heat wave days. In

fact, the WRF@15 km performed better than its 5 km counterpart, which further underlines that the grid resolution might play a less important role for this aspect. REMO is the RCM with the overall best performance due to the best values in all regards (Table 2), COSMO-CLM performed worst.

### 4.3 Heat wave characteristics

### 4.3.1 Heat wave frequency

Figure 4 shows the E-OBS pattern of the number of heat waves for the time period 1980 – 2009 and the grid cell based differences with the RCMs. The relevant scores are given in Table 3.

**Table 3.** Heat wave frequency metrics.

| Model | Mean [n heat waves] | Mean Bias [n heat waves] | SPAEF |
|---|---|---|---|
| COSMO-CLM | 29.7 | -1.16 | -0.14 |
| ALADIN 6.3 | 31.2 | 0.31 | -0.12 |
| REMO2015 | 30.5 | -0.33 | -0.04 |
| RegCM 4.6 | 33.4 | 2.57 | -0.24 |
| RACMO 2.2e | 31.2 | 0.38 | -0.22 |
| RCA4 | 29.2 | -1.71 | -0.25 |
| WRF@5 km | 26.7 | -4.14 | -0.25 |
| WRF@15 km | 29.5 | -1.37 | -0.25 |
| E-OBS | 30.9 | | |

The E-OBS domain looks mostly uniform with the majority of values ranging between 26 – 34 heat waves. The domain mean value (30.9) in Table 3 underlines this. Only in the north and south there are small concentrations of higher values up to 40 heat waves. The RCM bias domains show rather mixed patterns with both, positive and negative values. RegCM (positive) and WRF@5 km (negative) are the only domains where one direction in bias predominates. This is also reflected in the mean bias values in Table 3, where these two RCMs have the highest values (2.57 and -4.14), while at the other domains the opposite

signs tend to cancel each other out, bringing them closer to zero on average. The northern part of all bias domains is predominated by negative bias values. It is also noticeable that in the eastern part of the domain only positive values prevail in RegCM, while negative values prevail in all other bias domains in this area. The WRF@15 km experiment looks quite balanced and therefore quite different from its 5 km counterpart. This is also confirmed by the smaller mean bias value (-1.37) in Table 3. The domain mean values in Table 3 show a relatively large range of about seven heat waves between the maximum

(33.4 at RegCM) and minimum (26.7 at WRF@5 km) value. The E-OBS reference value of 30.9 means that in average there was approx. one summer heat wave per year in the study period. ALADIN and RACMO (31.2) come closest to this value, WRF@5 km (26.7) shows the biggest discrepancy. COSMO-CLM, REMO, RCA4 and the two WRF runs simulated in average less heat waves than the reference, ALADIN, RegCM and RACMO more. ALADIN has the lowest mean bias value (0.31). The mean bias values of COSMO-CLM, REMO, RCA4 and the two WRF runs are negative, the others are positive. All the

SPAEF scores between the reference and the single RCM domains (not shown) are negative here, indicating that there is no

good overall spatial agreement at all. The highest score (-0.04) is found at REMO, the lowest (-0.25) at RCA4 and the two WRF runs.

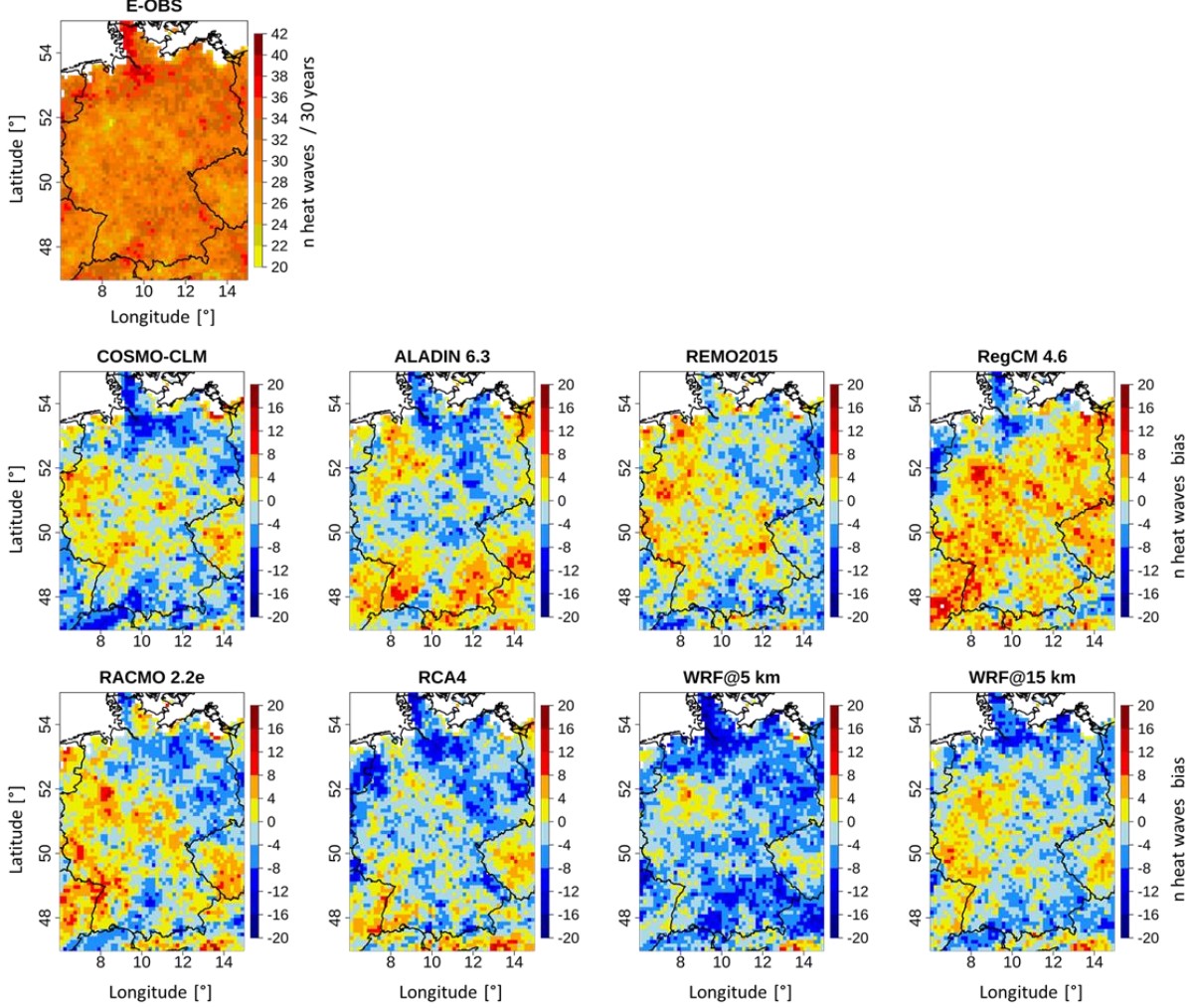

**Figure 4.** Grid cell based E-OBS summer heat wave frequency pattern between 1980–2009 and differences between each
RCM and E-OBS.

There are no recognizable benefits of the two WRF runs either. Here, it is rather the opposite, especially regarding the WRF@5 km run, since it shows the highest mean bias value and the biggest difference to the domain mean value in the number of heat waves compared to the reference. In addition, it has the lowest SPAEF value, which is not very meaningful in this case. The
model settings seem to have a clearly higher importance than the grid resolution. ALADIN showed the best performance in this section, closely followed by REMO.

### 4.3.2 Mean heat wave duration

Figure 5 displays the E-OBS pattern of the mean heat wave durations for the time period 1980 – 2009 and the grid cell based differences with the RCMs. The associated scores are shown in Table 4.


**Table 4.** Mean heat wave duration metrics.

| Model | Mean [n days] | Mean Bias [n days] | SPAEF |
|-------|---------------|--------------------|-------|
| COSMO-CLM | 5.46 | 1.53 | -0.65 |
| ALADIN 6.3 | 4.66 | 0.72 | -0.07 |
| REMO2015 | 4.17 | 0.23 | -0.13 |
| RegCM 4.6 | 4.50 | 0.56 | -0.38 |
| RACMO 2.2e | 4.14 | 0.20 | -0.21 |
| RCA4 | 4.37 | 0.43 | -0.24 |
| WRF@5 km | 3.78 | -0.16 | 0.06 |
| WRF@15 km | 3.84 | -0.10 | -0.15 |
| E-OBS | 3.94 | | |

The E-OBS pattern is very uniform, the majority of the domain is covered by values between 3.75 and 4.25 days. This is also reflected in the domain mean value of 3.94 days (Table 4). This means that the average heat wave duration was quite close to
the minimum length (3 days) of a heat wave. All EURO-CORDEX RCM bias domains except REMO and RACMO are prevailed by positive bias values, meaning that the models simulated too long heat episodes. The COSMO-CLM domain even appears to be without any negative bias grid cell. It is also the domain with the highest values. Especially the area in the southwest, where values up to 5 days are reached, is striking. All other bias domains are predominated by values between -1 and 1 day. The two WRF outputs are the only ones prevailed by negative bias values. In this case, the WRF@15 km domain
is quite close to its 5 km counterpart, but again the negative bias is less pronounced in direct comparison. The patterns of REMO and RACMO are similar. The domain mean values in Table 4 are all close to each other. All EURO-CORDEX RCMs are above the reference value (3.94 days), the WRF runs are below. COSMO-CLM shows the biggest difference (1.52 days), WRF@15 km the smallest (0.1 days). COSMO-CLM is also the RCM with by far the highest mean bias value (1.53 days), which was to be expected from the bias maps (Figure 5). It is the only case where the mean bias is greater than 1 day. The bias
value of 1.53 days may seem relatively small, but if it is set in relation to the domain mean values, then it accounts for a fairly large proportion. Only the two WRF runs have negative mean bias values. WRF@15 km holds the smallest (-0.10) mean bias value. Here, the SPAEF values between the reference and the single RCM domains (not shown) are all negative or very low, which is the case for WRF@5 km (0.06). This means that again no RCM was able to satisfactorily reproduce the spatial pattern of the reference. COSMO-CLM holds also in this regard the worst value (-0.65).

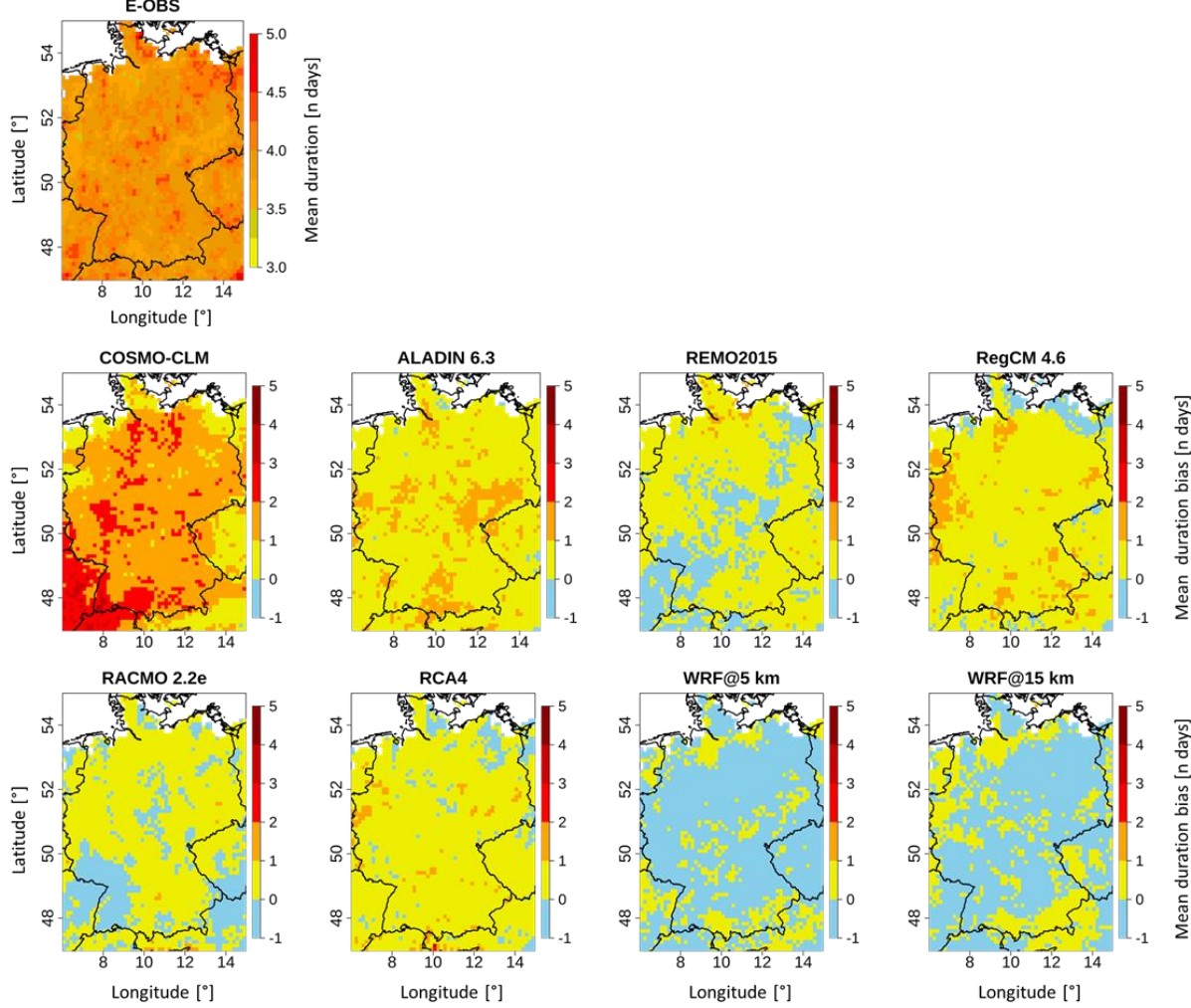


**Figure 5.** Grid cell based E-OBS summer mean heat wave duration pattern between 1980–2009 and differences between each RCM and E-OBS.

COSMO-CLM is clearly the RCM with the weakest performance due to the weakest scores in each aspect (Table 4). WRF@15

km is the most reliable RCM here, meaning that there are indeed some benefits for this aspect of the analysis, which are rather related to the model settings, since once again, the 15 km performs better than its 5 km counterpart.

### 4.3.3 Mean heat wave intensity

Figure 6 provides the E-OBS pattern of the mean heat wave intensity for the time period 1980 – 2009 based on the cumulative heat measure. The grid cell based differences of the RCMs are also included. The relevant scores are given in Table 5.

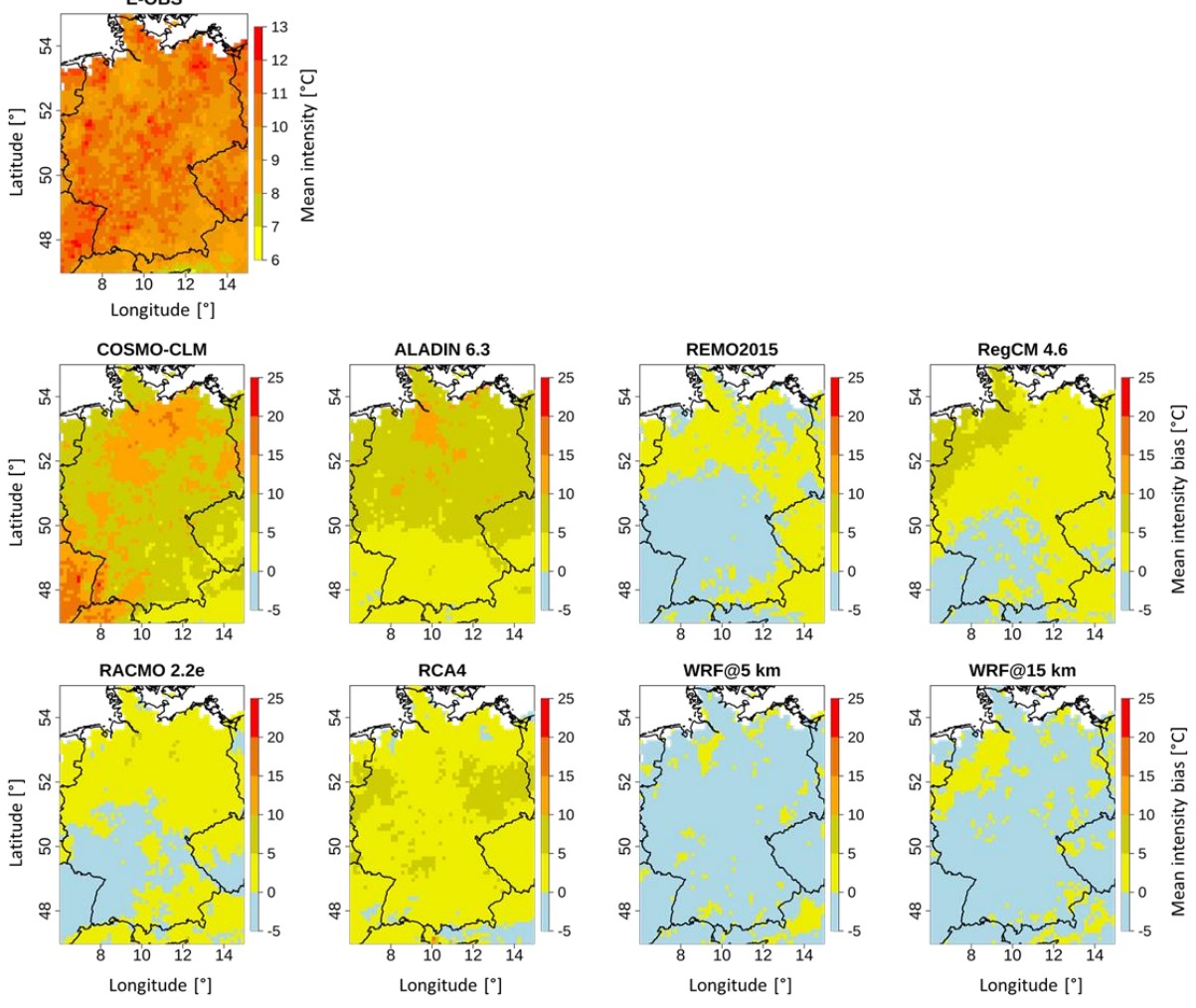


**Figure 6.** Grid cell based E-OBS summer mean heat wave intensity pattern between 1980–2009 and differences between each RCM and E-OBS.

**Table 5.** Mean heat wave intensity metrics.

| Model | Mean [°C] | Mean Bias [°C] | SPAEF |
|-------|-----------|----------------|-------|
| COSMO-CLM | 18.9 | 8.88 | -0.52 |
| ALADIN 6.3 | 15.6 | 5.66 | -0.48 |
| REMO2015 | 9.8 | -0.17 | -0.19 |
| RegCM 4.6 | 11.7 | 1.73 | -0.85 |
| RACMO 2.2e | 10.9 | 0.90 | -0.49 |

| | | | |
|---|---|---|---|
| RCA4 | 13 | 3.04 | -0.40 |
| WRF@5 km | 8.8 | -1.17 | 0.03 |
| WRF@15 km | 9.12 | -0.85 | 0.11 |
| E-OBS | 9.97 | | |


The E-OBS domain looks quite uniform. A sort of band of higher values extends from the southwest to the northeast. The majority of the values lies within 9 to 11 °C, which is confirmed by the domain mean value of 9.97 °C (Table 5). Accounting for the mean duration (3.94 days) of heat waves from the section above, the average heat excess per day during a heat wave period was 2.53 °C. Regarding the RCM bias maps, the two WRF simulations are again predominated by negative values and

look quite similar. Areas of positive bias in the 15 km domain are similarly situated as in the 5 km counterpart. Some of them are more extensive like in northwestern part, others are smaller like in southeast. The domains of COSMO-CLM, ALADIN and RCA4 are prevailed by positive values, while those of REMO, RegCM and RACMO show a mixed pattern. In those domains, the areas of negative bias are similar between the models, mainly located in the southwest. They also have in common that the values are mostly between -5 to 5 °C. This leads to relatively small mean bias values (Table 5) due to mutual balancing.

Maximum bias values of up to 25 °C are found at the COSMO-CLM domain in the southwestern part. The COSMO-CLM and ALADIN domains are basically covered with comparatively higher values. The domain mean values in Table 5 show distinct differences among themselves with a maximum range of 10.1 °C between COSMO-CLM and WRF@5 km. The two WRF runs and REMO are below the reference value (9.97 °C), all other models are above. REMO's value (9.8 °C) comes closest to the reference, COSMO-CLM (18.9 °C) shows by far the largest difference. Its value is almost double as high as the reference.

As inferred from the maps, REMO and the two WRF runs have negative mean bias values, while the other RCMs have positive ones. REMO holds the smallest value (-0.17 °C), COSMO-CLM the highest (8.88 °C), which, compared with the domain mean values, is a very high value. The SPAEF values between the reference and each RCM domain (not shown) are once more all negative (all EURO-CORDEX RCMs) or very low (the two WRF runs), which means that there is no satisfactory reproduction of the reference's spatial structure. RegCM has the lowest value (-0.85), WRF@15 km the highest (0.11).

COSMO-CLM is the weakest performing RCM, while REMO showed the overall best performance. This means that there are no real WRF benefits apparent here, except for the possible minor benefits in reproducing the spatial structure. It is striking that the pattern of the WRF domains always being prevailed by negative bias runs through all aspects of the heat wave characteristics as well as the number of heat wave days.

## 4.4 Heat wave trends

Figure 7 presents the grid cell based results of the Mann-Kendall trend test for the annual number of heat waves in the study period 1980 – 2009 for all RCMs and the E-OBS reference. A summary for each signal and data set is given in Table 6. In this context it should be noted that Mann-Kendall trend test provides information about whether there is a monotonic positive,

negative or no trend in a time series at a certain level of significance (here 0.05). It does not give information about exact trend values.


**Table 6.** Annual number of summer heat wave overall metrics.

| Model | negative [%] | neutral [%] | positive [%] |
|---|---|---|---|
| E-OBS | 0 | 88 | 12 |
| COSMO-CLM | 0 | 99.74 | 0.26 |
| ALADIN 6.3 | 0 | 99.97 | 0.03 |
| REMO2015 | 0 | 99.56 | 0.44 |
| RegCM 4.6 | 0 | 90.35 | 9.65 |
| RACMO 2.2e | 0.03 | 99.95 | 0.03 |
| RCA4 | 0 | 95.46 | 4.54 |
| WRF@5 km | 0 | 96.87 | 3.13 |
| WRF@15 km | 0 | 96.09 | 3.91 |

The domains of COSMO-CLM, ALADIN, REMO and RACMO are almost completely covered with no trend signals. This also confirmed by the values in Table 6, where each of these RCMs have more than 99 % of their grid cells in the neutral

section. RegCM and RCA4 have distinct areas of positive trends, but in different areas. WRF@5 km has also positive trend areas, but less concentrated. In the E-OBS reference domain there are also concentrated areas of positive trends, mainly in the southwest and in the northern central area. These locations mostly do not agree with those from the RCMs. The WRF@15 km domain also shows positive trend grid cells, but at different areas than the other RCMs. It shows the highest spatial agreement with the reference, especially in the central part. It is striking that there are actually no grid cells with negative i.e. decreasing

trend in any domain, which is also confirmed in Table 6. The table further confirms that the majority of the grid cells is covered by no trend signals. By far the largest proportion of positive grid cells can be found in E-OBS (12 %), followed by RegCM (9.65 %), RCA4 (4.54 %) and the WRF runs (3.13 % and 3.91 %). All other runs are in the less than 1 % range. This shows that the WRF@15 km run is not only from the locations, but also from the shares of the signals closer to the reference than its 5 km counterpart.

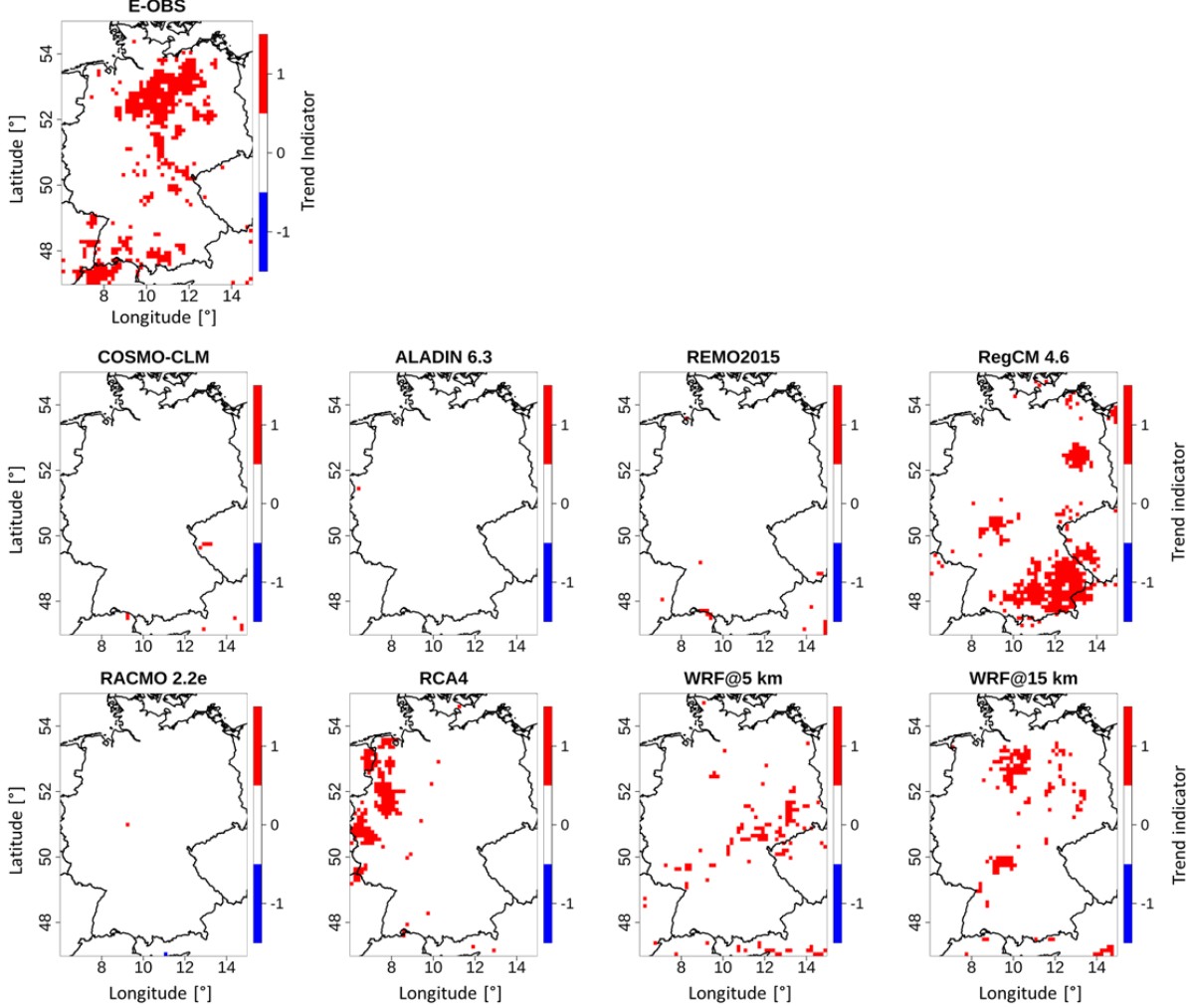


**Figure 7.** Grid cell based trends of the number of annual summer heat waves for 1980–2009 based on the Mann-Kendall Test for E-OBS and each RCM.

This section reveals that, according to the E-OBS reference, if there is a trend in number of heat waves, it is only positive,

meaning that the frequency is increasing with time. But this is not the case everywhere. All models simulate too few pixels with positive trends. Regarding WRF, any possible benefits would be related to the model settings rather than to the grid resolution, since the WRF@15 km is more accurate than its 5 km counterpart.

## 4.5 The heat wave event 2003

### 4.5.1 Cumulative heat

Figure 8 shows the E-OBS cumulative heat pattern of the summer season 2003 and the grid cell based differences with each RCM. The associated scores are given in Table 7.

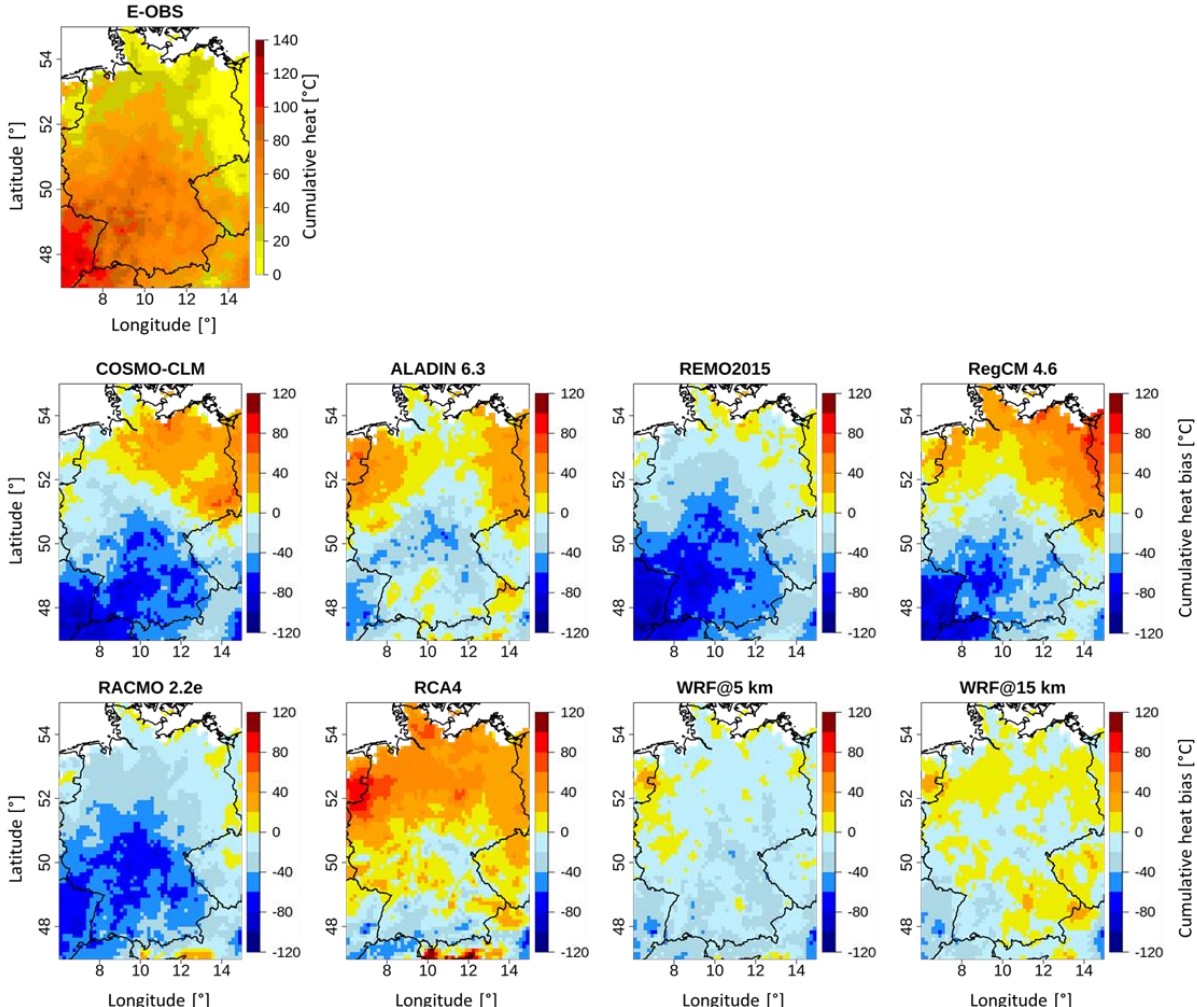

**Figure 8.** Grid cell based E-OBS summer 2003 cumulative heat pattern and differences between each RCM and E-OBS.

The E-OBS domain shows a quite clear gradient from the southwest to the northeast with decreasing values. The values in the
southwest are very high, they have accumulated well above 100 °C during that summer season, making the heat wave most pronounced in this region. These values underline the high intensity of the 2003 heat wave. The mildest values of up to 10 °C heat excess are in the northeastern regions, making them the regions least impacted by the heat wave. COSMO-CLM, REMO, RegCM and RACMO show pronounced negative bias values in the southwest of their domains. This means that they were not

able to satisfactorily reproduce the particularly high values of the reference in these regions, showing bias values of up to -120

°C. These bias values indicate that the models simulated only weak or even no heat episodes at all in regions, where the reference showed the most pronounced values. REMO, RACMO and WRF@5 km are prevailed by negative bias values, the remaining RCMs except WRF@15 show mixed patterns, where the northern half is prevailed by positive, the southern half by negative bias, leading to a sort of bipartition. The WRF@15 km is evenly covered with values of both signs. The WRF@5 km pattern is most uniform with relatively low bias values all over the domain. It is striking that in contrast to the previous sections,

the WRF domains are not the only ones prevailed by negative bias values here. In direct comparison with its 5 km counterpart, the WRF@15 km domain has much more areas with positive bias. The domain mean values in Table 7 show a large discrepancy among each other with a range of 52.7 °C between RCA4 (62.3 °C) and REMO (9.6 °C). The reference value of E-OBS is 45.4 °C. Considering the mean heat wave intensity (9.9 °C) for the whole study period from Sect. 4.3.3, this value is more than four times higher and thus very remarkable and it illustrates the great severity of this heat wave. ALADIN (43.2 °C) is closest

to the reference value, REMO holds the largest difference. Regarding the mean bias values, REMO (-35.9 °C) and RACMO (-34 °C) have the highest values, ALADIN (-2.2 °C) the lowest. It needs to be considered that in the RCMs with the bipartition pattern mentioned above (COSMO-CLM, ALADIN and RegCM), the values cancel each other out, leading to relatively low mean bias values, depending on the degree of balance. Only RCA4 holds a positive mean bias value (16.9 °C). This underlines that the models rather underestimate the intensity of this heat wave period. The WRF@5 km mean bias (-12.9 °C) is clearly

higher than that of its 15 km counterpart (-3.4 °C). Here there are some distinct differences between the SPAEF values. While they are negative for the most EURO-CORDEX RCMs or very low (ALADIN and RCA4), the two WRF runs show relatively high values (0.77 for WRF@5 km and 0.72 for WRF@15 km). This means that the WRF runs reproduced the spatial structure of the reference reasonably well. RegCM holds the lowest (-0.69) SPAEF value.

**Table 7.** Cumulative heat 2003 metrics.

| Model | Mean [°C] | Mean Bias [°C] | SPAEF |
|-------|-----------|----------------|-------|
| COSMO-CLM | 25.6 | -19.8 | -0.38 |
| ALADIN 6.3 | 43.2 | -2.2 | 0.26 |
| REMO2015 | 9.6 | -35.9 | -0.04 |
| RegCM 4.6 | 35.9 | -9.5 | -0.69 |
| RACMO 2.2e | 11.4 | -34 | -0.03 |
| RCA4 | 62.3 | 16.9 | 0.09 |
| WRF@5 km | 32.5 | -12.9 | 0.77 |
| WRF@15 km | 42.1 | -3.4 | 0.72 |
| E-OBS | 45.4 | | |

There are distinct differences between the single models in this section. ALADIN is the RCM with the overall best performance, REMO the one with the worst. There are pronounced benefits of the WRF runs visible in the reproduction of the spatial structure of the reference. In this regard, the 5 km WRF run performs slightly better than its 15 km counterpart. In terms
of reproducing the reference domain mean value and of mean bias value, the 15 km WRF run outperforms its 5 km counterpart.

### 4.5.2 Maximum duration

Figure 9 gives the E-OBS pattern of the maximum heat wave duration during the 2003 summer season along with the grid cell based differences with each RCM. The corresponding values are given in Table 8.

**Table 8.** Maximum heat wave duration 2003 metrics.

| Model | Mean [n days] | Mean Bias [n days] | SPAEF |
|---|---|---|---|
| COSMO-CLM | 6.91 | -1.76 | -0.64 |
| ALADIN 6.3 | 6.34 | -2.33 | -0.17 |
| REMO2015 | 4.30 | -4.37 | -0.42 |
| RegCM 4.6 | 5.51 | -3.15 | -0.69 |
| RACMO 2.2e | 2.87 | -5.80 | -0.44 |
| RCA4 | 8.94 | 0.27 | -0.35 |
| WRF@5 km | 7.72 | -0.95 | 0.19 |
| WRF@15 km | 8.86 | 0.19 | 0.07 |
| E-OBS | 8.67 | | |

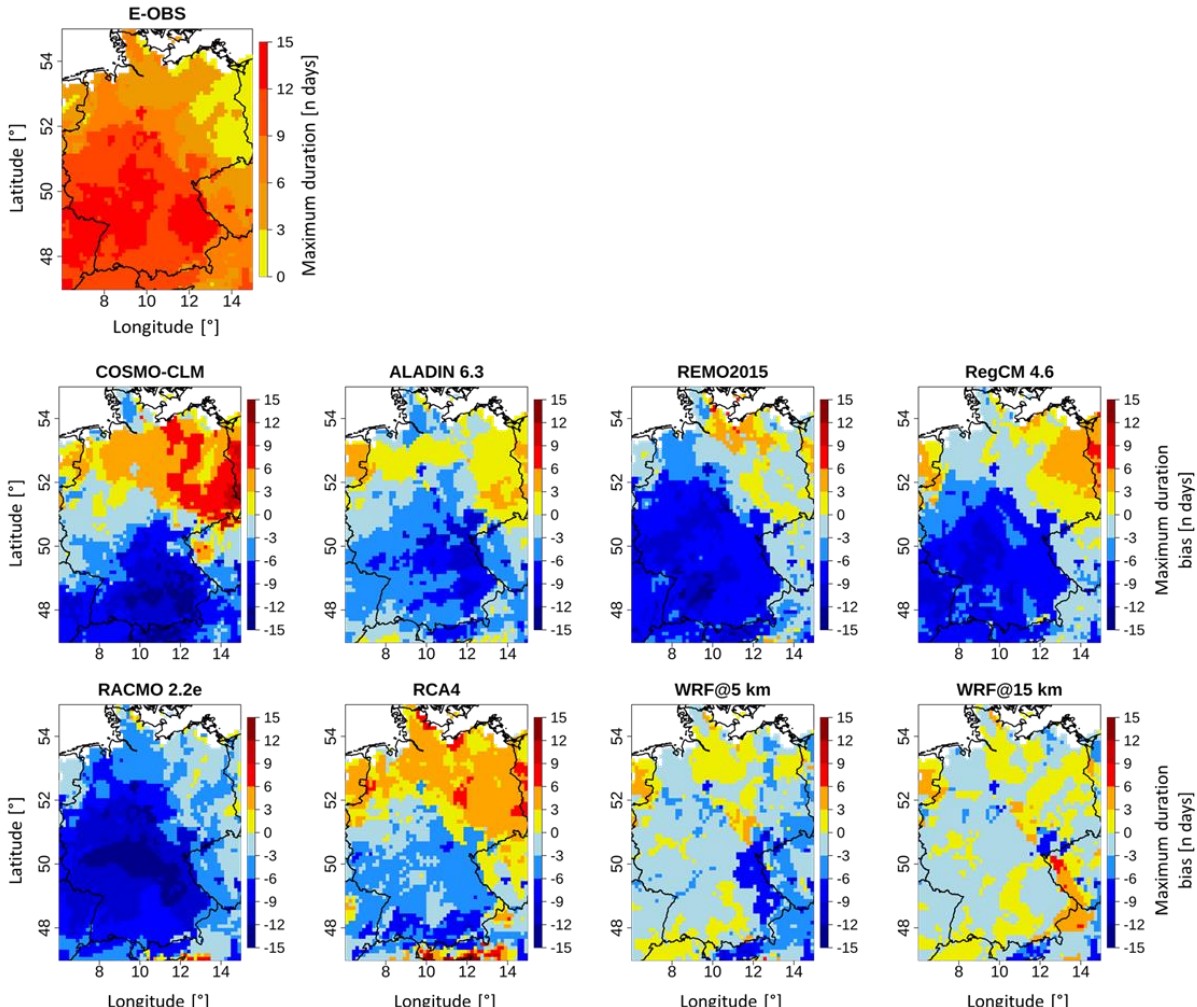

**Figure 9.** Grid cell based E-OBS summer 2003 heat wave maximum duration pattern and differences between each RCM and E-OBS.

The E-OBS domain shows a sort of bipartition with the highest values in the southwestern and lower values in the northeastern part. This matches the impressions from the section above, where the highest heat excess values were also found in the southwest (Figure 8). Due to the longest durations of up to 16 days, the high heat excess values could accumulate. It should be noted that this is not only a matter of duration, but also of excess values. In the northeastern part there are areas with values ranging between 0 – 3. Since the minimum duration of a heat wave episode was defined to be three days, it means that in these regions no heat episode took place in that summer period. This is also in line with the heat excess pattern (Figure 8), which shows values between 0 and 10 °C in some of these areas. The RCM bias patterns roughly match those in Figure 8. COSMO-CLM, REMO, RegCM and RACMO have strong negative bias values of up to -15 days in the southern and southwestern areas. Since the E-OBS domain shows the highest values of up to 15 days in some parts of those areas, it further confirms that the

models actually did not simulate a heat wave episode in some parts of these areas, where the reference shows the most distinct

values. It further underlines this big shortcoming of the models. The northern parts in COSMO-CLM and RCA4 are prevailed by positive bias values. COSMO-CLM is the model with the largest areas of high positive bias values of up to 15 days in the eastern parts. This is the region where the E-OBS reference only shows weak or even no heat wave episodes, meaning that the model does the opposite here compared to the southwestern region, simulating relatively strong heat episodes although there were none according to reference. In almost each domain there are high bias values in the Alps region in the south. With the

exception of RCA4, these bias values are all negative. In line with the section above, the WRF domains are not the only ones prevailed by negative values. In the southwestern regions, they both show the lowest bias values of all RCMs. The WRF@15 km domain shows the most balanced pattern between positive and negative bias values with most of them ranking in the relatively low range. This is confirmed by the lowest mean bias values (0.19 days) in Table 8. The domain mean values in Table 8 reveal that the E-OBS reference value (8.67 days) is one of the highest, only exceeded by RCA4 (8.94 days) and

WRF@15 km (8.86 days). This is also reflected in the mean bias values. It indicates that the models tend to simulate shorter durations. The WRF@15 km value is also the closest to the reference, while RACMO (2.87 days) shows the biggest difference. It is also by far the lowest value. The model also holds the highest mean bias value (-5.80 days). It needs to be remembered that the low mean bias values of WRF@15 km and RCA4 (0.27 days) also result from the contradictory nature of their values. Analogous to the cumulative heat, the two WRF runs hold the best SPAEF values (0.19 and 0.07, respectively), but on a much

lower level, making them not very meaningful. The score of the WRF@5 km experiment is clearly better than that of its 15 km counterpart. All SPAEF values of the EURO-CORDEX RCMs are negative. The lowest score is found at RegCM (-0.69). Overall, no model achieves to reproduce the spatial structures satisfactorily.

According to the scores, WRF@15 km shows the best overall performance in this section, RACMO the worst. This means that once more WRF@15 km outperformed its 5 km counterpart. The great weakness of COSMO-CLM of simulating oppositely

needs also to be considered, since this may not be reflected in the values of the table. The fact that the WRF runs, especially the 5 km run, in this as well as in the previous section were not the only models prevailed by negative bias values (Figure 8 and Figure 9) like in all the previous sections above, highlights that for a single event, the situation can be very different compared with the overall picture for the complete study period provided in the Sect. 4.2 and 4.3.

## 5. Discussion

Regarding the $T_{max}$ reproduction in Sect. 4.1, Silva et al. (2022), who compared monthly $T_{max}$ values from six historical runs from different GCM-RCM combinations from CORDEX-CORE with ERA5 as reference for the Pantanal region for the period April - October between 1981 and 2005, found temporal, area averaged correlation values between 0.42 and 0.67. This is distinctly less than in our case. The different RCM forcings and focus on different period and region must be considered though. As for the bias values (Table 1), in previous studies, a negative bias of daily $T_{max}$ was found for the Central European

region (Nikulin et al., 2011; Plavcová and Kyselý, 2011). Here, we found both, positive and negative bias, depending on the RCM.

Regarding the heat wave characteristics (Sect. 4.3), there are different reasons discussed in the literature for the over and underestimation by the models. Lhotka et al. (2018b) assume that the overestimation of heat wave frequency and duration of major heat waves, which they found in some RCMs, is related to the large-scale circulation and soil moisture depletion. Underestimation of these events, on the other hand, is associated with too-moist summertime conditions. Vautard et al. (2013), who, like in this study, also found that simulated heat waves from the EURO-CORDEX RCMs were too long and intense (not the case for REMO here), attribute this to biases in the modeled temperature. They state that there "is no clear explanation" for these biases. They suspect that overestimation of heat waves is connected to the combination of anticyclonic weather and amplifying land–atmosphere feedback. Exaggeration of land-atmosphere feedback could lead to asymmetry and skewness in the temperature distribution (Jaeger and Seneviratne, 2011), which can stretch temperature values at the extremes and in turn induce higher amplitudes and durations of events. It must be noted that they used the daily mean instead of maximum temperature for their heat wave definition. Lhotka and Kyselý (2015a) also go in the direction of land-atmosphere feedback, since they found a connection between heat wave intensity and precipitation during and before these events. Vautard et al. (2013) further found that coarser resolution lead to very persistent heat waves. This is also how it looks here at first, if only the WRF@5 km run is compared with the EURO-CORDEX RCMs. The WRF@15 km run shows that this cannot be related to resolution, but to setting effects, since it is coarser then the EURO-CORDEX runs. It must be noted that the resolution differences in Vautard et al. (2013) (50 km vs. 12.5 km) were much more distinct than in our case. According to Plavcová and Kyselý (2019), an overestimation of circulation super types may contribute to the development of too-long heat waves in some simulations.

To investigate potential sources of the bias, we additionally analyzed the following variables from the EURO-CORDEX outputs (the corresponding figures can be found in the supplement): sensible and latent heat flux, incoming and outgoing short and long wave radiation, soil moisture and surface pressure. Other outputs from the WRF experiments were no longer available. We performed a correlation analysis (Figure S1 and S3) to identify dependencies among the individual variables for each model with focus on $T_{max}$ dependencies. Across all models, highest $T_{max}$ correlations were found with the radiation variables, especially with the longwave outgoing radiation. In the next step we compared the distributions of the single variables among the individual models from boxplots (Figure S2 and S4). No clear conclusions could be drawn, since the distributions of the variables that show highest $T_{max}$ correlations are quite similar among the models. Exceptions are the sensible heat flux, which does not show a high $T_{max}$ correlation, and soil moisture. Comparison of soil moisture between models is not considered useful, since there are considerable differences in the modeled soils, like different number of soil layers (e.g., five layers in ALADIN, three layers in RCA4 and four layers in WRF), different layer depths etc. Anyways, soil moisture does not show high correlations with $T_{max}$ either. The described procedure was conducted for both, the summer months of the entire study period and the summer months of 2003 to ensure that heat wave conditions are included in the analysis. Since the temporal course of the variables is decisive and the consideration of the distributions allows only little statements about this, we have additionally

looked at the spatially averaged courses of the individual variables for each model run for the summer 2003 months (Figure

S5). Largest differences between the individual models are found for the soil moisture here, too, which is not very meaningful, as already mentioned. Naturally, there is a spread between the single lines for each variable, but they greatly agree in their patterns or variability, respectively, which is more important than the agreement in the actual values. Thus, the differences that occur partly, especially for the radiation variables, since they show highest correlations, may provide explanation only to some degree, since it is not possible to establish overall consistency with the results. COSMO-CLM for example, that was shown to

overestimate mean heat wave duration and severity the most (Sect. 4.3.2 and 4.3.3), does not stand out in this view. Additional potential reasons for bias like different land use data between the single models was beyond our ability to investigate. From an overall perspective, the heat wave characteristics are generally quite well captured in terms of spatiotemporal mean values (Table 3 - 5). Lin et al. (2022) came to similar findings, even though they used different heat wave metrics.

As for the heat wave event 2003 (Sect. 4.5), Lin et al. (2022) associate the high bias values in the Alps region for the maximum

durations with the representation of orographic effects. Russo et al. (2016) also found a big discrepancy in the RCM's capability to simulate a single major heat wave event. In fact, only one out of 13 RCMs was able to capture the event. They attribute this to some model deficiency simulating really extreme heat waves or to the length of the analyzed time period (1979 – 2005). This underlines that increased resolution does not lead to improved reproduction of severe heat waves in most cases. Lhotka et al. (2018a) as an exception could find exactly that. It must be considered that the difference between the resolutions they

compared (12.5 vs. 50 km) was much larger than in our case.

In this context the role of model internal variability should also be discussed. It has been found to depend on the variable, the season and the domain size. In the midlatitudes it is higher summer and smaller in winter. The boundary forcing is weaker in summer time compared to winter, so that the model is freer to develop its own internal dynamics (Caya and Biner, 2004; Lucas-Picher et al., 2008; Lavin-Gullon et al., 2020). Additionally, during winter the westerly flow is stronger, sweeping away

internally generated model response to the forcing (Giorgi and Bi, 2000). Since only summer periods are regarded in this study, this possibly plays a role for the model performances. Furthermore, the internal variability was shown to play a bigger role for precipitation than for temperature (Giorgio and Bi, 2000; Laux et al., 2017; Lavin-Gullon et al., 2020; Yu et al., 2020). Since it is all about temperature in this study, this fact points towards a smaller role of internal variability. Moreover, it was found that smaller domain sizes are associated with lower internal variability (Giorgi and Bi, 2000; Rinke and Dethloff, 2000;

Vanitsem and Chomé, 2005; Alexandru et al., 2007; Lucas-Picher et al., 2008; Lavin-Gullon et al., 2020). This is because in larger domains the lateral boundary control is reduced due to the large area, so that the RCMs have more freedom to develop their own characteristics (Lucas-Picher et al., 2008). This likely plays a role in this study, since there is a crucial difference in the domain sizes between the EURO-CORDEX RCMs and the WRF experiments. The EURO-CORDEX domain is far bigger than the second domain of the WRF experiment. This increases the potential for internal variability in the EURO-CORDEX

runs compared to the WRF experiments. However, since these do not show significantly better performance, the role of internal variability rather seems limited. Internal variability is often associated with the successful reproduction of single events. It is known for being a reason why finding or reproducing one particular extreme event, which is by definition a rare event, of the

same magnitude, duration, spatial scale and location as from the observations should not be expected (Jain et al., 2023). In this case it relates to the heat wave event of summer 2003, where the models were shown to struggle with accurate reproduction (Sect. 4.5). Some part of these struggles may be explained by internal variability. However, it needs to be considered that the entire summer period of 2003 was considered with its individual heat episodes, as described in Sect. 3.5, which should reduce internal variability's role. For the reproduction of the heat wave characteristics in Sect. 4.3 the summer months for the entire study period were considered, so that the values in the Table 3–5 give the overall average values. This should considerably reduce the role of internal variability, as it was shown to not affect the domain-wide average climatology (Giorgio and Bi, 2000). The internal variability further depends on the model configuration (Giorgi and Bi, 2000), so that in this case it likely varies depending on the model.

In each section of this study we found an inter-model spread. This is in line with previous heat wave related model employing studies, e.g., Vautard et al. (2013), Gibson et al. (2017), Feron et al. (2019) and Silva et al. (2022). Vautard et al. (2013), who also used ERA-Interim driven EURO-CORDEX outputs, assume several potential sources of spread: the method to process boundary conditions in the model, the convection treatment, the different parameterizations, which are more pronounced in larger domain sizes and specific weather situations and the way in which the interactions between land surfaces and the atmosphere are accounted for in the models. The latter refers to the uncertainty of partitioning between sensible and latent heat flux as well as of radiation fluxes (Lenderink et al., 2007; Stegehuis et al., 2012).

In each of the heat wave related sections, no evidence is found that increased resolution leads to better results in reproducing the related metrics. In fact, the WRF@15 km always performed better than its 5 km counterpart. This is partly in line with the results of previous studies. No benefits of increased resolution were also found in Plavcová and Kyselý (2019), who compared 25 km and 50 km resolution and in Molina et al. (2020) (12.5 km vs. 50 km). Cardoso et al. (2019), who also compared 12.5 and 50 km resolution, found a slight benefit of increased resolution. Careto et al. (2022) and Lin et al. (2022) compared 0.11° RCM outputs with the outputs of the driving data in much higher, different resolutions. In both studies benefits of the increased resolution were found, particularly for coastal regions. Vautard et al. (2013) (12.5 vs 50 km) identified partly benefits of increased resolution, depending on the aspects of analysis. They found that increased resolution leads to reduction in biases in the temperature 90th percentile as well as in the heat wave persistence. They suspect that smaller scale and higher frequency variability are better resolved in higher resolutions. Local bias improvements were found in some coastal regions here too. Once more it is noted that in each of the mentioned studies, the difference between the employed resolutions is much higher than in our case here. We assume from our results that a resolution increase from an already relatively high resolution, in this case 12.5 km, has limited to negligible impact. One fact that also needs to be considered is the original resolution difference between E-OBS (12.5 km) and WRF@5 km. For certain aspects, structures from E-OBS may be better represented in resolutions closer to it. This could be a reason why the 15 km WRF runs performs better with otherwise the same settings. Furthermore, it needs to be kept in mind that in the E-OBS data set extreme values tend to be smoothed out due to interpolation processes (Haylock et al., 2008; Hofstra et al., 2009). This can mean that for certain aspects of the analysis, where the models

showed negative bias, discrepancy to the true values might be even bigger. Hofstra et al. (2009) emphasize, that this effect is more pronounced for precipitation though.

The three RCMs with the overall best performances are ALADIN, REMO and WRF@15 km. COSMO-CLM showed the weakest overall performance. If this is considered, the choice of the parameterization schemes seems to play a minor role, since ALADIN, REMO and WRF have all different schemes for the individual physics. This finding is opposite to that from Davin et al. (2016) who identified the land surface scheme as highly important for a proper simulation of temperature. The land surface scheme is determining for two crucial factors: the soil moisture and leaf area index (LAI). If the LAI within the land-surface model is based on climatological values instead of dynamical calculations, this can increase evapotranspiration and thus lead to a cooling effect, which reduces the maximum temperature values. The connection between adjacent soil moisture conditions and heat extremes is well documented (e.g., Brabson et al., 2005; Fischer et al., 2007; Lorenz et al., 2010; Hirschi et al., 2011; Jaeger and Seneviratne, 2011; Liu et al., 2013). One possible determiner for bias can further be the microphysics scheme, which is responsible for the cloud processes. In earlier studies, the role of cloud cover for the (maximum) temperature simulation is highlighted. An increased cloud cover leads to a greater fraction of reflected solar radiation, which in turn leads to cooling of $T_{max}$ (Groisman et al., 2000; Sun et al., 2000). Lobell et al. (2007) found that cloud cover is responsible for higher daily $T_{max}$ variability compared to the daily mean values. They consider the cloud cover especially important during the summer period. According to Liang et al. (2008), biases in simulated radiation budgets can lead to errors in surface temperatures. Hamdi et al. (2012) found strong correlations of positive bias with cloud cover representation. However, since all models in this study were ran with different microphysics schemes (Table 2 in Petrovic et al. (2022)), relevant conclusions cannot really be drawn. Interestingly, in the study of Lhotka et al. (2018b), COSMO-CLM in combination with a driving GCM was among the RCMs with the best performances in simulating major European heat waves. This could indicate the high importance of the driving data, which is also highlighted by Molina et al. (2020). It is further underlined by the fact that RACMO and RCA4, driven by the same GCM, also showed best performances, while in our study they did not stand out. It is evident that the WRF runs, compared to the EURO-CORDEX RCMs, often showed rather negative bias values. This might be connected with the employed schemes in the WRF simulation (Table 2 in Petrovic et al. (2022)), since there are no commonalities with the other runs at the individual physics. Moreover, a connection between the $T_{max}$ bias (Table 1) and the overall performance does not seem to exist, since COSMO-CLM has the lowest mean bias value (-0.16 °C), but the worst overall performance, while on the other hand RCA4, which holds the highest mean bias value (-2.40 °C), does not show a significant bad performance. There are model output employing heat wave studies in which it has been found that the ensemble mean outperforms the individual models runs, e.g., Russo et al. (2016), Wang et al. (2019a), Lin et al. (2022) and Kim et al. (2023). Here, ensemble mean values were not considered since a focus of the study is about the impact of different model resolution and settings and evaluation of single model performances.

According to Plavcová and Kyselý (2019), biases in the simulations of atmospheric circulation play a crucial role for the simulation of temperature extremes. This is why they claim that an improvement in this field will be among the most important steps towards better reproduction of extreme temperature events and thus also lead to more credibility of future projections.

Model forcing also plays a role for the model's performance. As described above, in this case all model runs had the same ERA-Interim reanalysis forcing. In 2018, first parts of the successor of ERA-Interim, ERA5, were released by the ECMWF (Hersbach et al., 2020). ERA5 has a finer spatial and temporal resolution, uses a more advanced assimilation system and includes more data sources. This raises the question of whether model performance could be improved by ERA5 forcing. At the time of selection of the data sets used in this study and also to date, there are no reanalysis runs driven by ERA5 retrievable

from the EURO-CORDEX platform. In the recent past there have been a few studies dealing with the comparison of ERA-Interim and ERA5. These studies focus on comparison of certain variables such as precipitation (e.g., Nogueira, 2020; Lavers et al., 2022; Steinkopf and Engelbrecht, 2022), several variables including precipitation and temperature (e.g., Rakhmatova et al., 2021; King et al., 2022; Nacar et al., 2022) and cloud cover (e.g., Lei et al., 2020) for different parts of the world. Moreover, there are comparison studies regarding transport simulation (Hoffmann et al., 2019), impact on certain models (Albergel et al.,

2018), impacts on hydrological modeling (Tarek et al., 2020) and on atmospheric corrections for InSAR (Zhang et al., 2022). In most of the cases, ERA5 significantly performs better than ERA-Interim or contributes to better results. This is especially the case for the reproduction of precipitation. Thus, it is to be expected that the RCM outputs would benefit from an ERA5 forcing, leading to better performances. Due to the improvement especially in precipitation, benefits would likely be more relevant for drought than for heat wave analysis, especially since patterns are better reproduced (Lavers et al., 2022). It may

be worth to directly compare simulation outputs from the same RCM, driven by ERA-Interim and ERA5. This remains subject to future studies.

It is striking that there are some significant differences in the outcomes compared to the drought study (Petrovic et al., 2022), in which the same data sources were used as mentioned above. While it was found that all models performed at similar levels for the drought characteristics, there are some significant differences between the individual performances here for the heat

wave characteristics, highlighting that the choice of model can be crucial. In addition, it was shown that the WRF settings and increased resolution were particularly beneficial for reproducing drought trends. This is not the case for heat wave trends. This suggests that these benefits are highly related to the simulation of precipitation, the most important variable for drought, which is not a factor here. The different time scales could also be a factor. Unlike heat waves, where the minimum duration in this study is three days, droughts are prolonged events with a minimum duration of usually at least one month. For this reason,

monthly values were considered in the drought study, while daily values were used here. Regarding the reproduction of the 2003 drought and heat wave event, there are pronounced differences between the models in both studies. Interestingly, REMO shows the worst performance in this respect in both cases and significantly underestimates the drought and heat wave conditions, respectively.

## 6. Conclusions

A heat wave analysis for Germany and its near surroundings for the period 1980-2009 was performed. The impact of increased model resolution and appropriate model configuration on the reproduction of heat wave metrics based on the $T_{max}$ simulation

is addressed. For this purpose, we employ an ensemble of six ERA-Interim-driven EURO-CORDEX RCMs of 12.5 km horizontal grid resolution and an ERA Interim-driven high-resolution (5 km) WRF run, whose setup was tailored to the target area. The model outputs are evaluated with regard to their ability to reproduce $T_{max}$ and heat wave characteristics based on it, trends and the major event in 2003. E-OBS data is used as reference.

WRF with its increased resolution and tailored model settings is shown to be not necessarily beneficial for improved performance in reproducing heat indices. Benefits are somewhat present only for the reproduction of the mean heat wave durations and for the reproduction of the spatial structures of the cumulative heat values for the heat wave event 2003. In fact, the WRF@15 km run outperforms its 5 km counterpart in each section. Thus, we conclude that (for the selected model configurations) increased resolution does not contribute to better performances regarding heat wave metrics, when both compared resolutions are already relatively high, which is the case here (12.5 km vs. 5 km). Since three models, namely ALADIN, REMO and WRF@15 km show the overall best performances, we further conclude that the tailored model settings of WRF only have limited benefits for the reproduction of the heat wave metrics. The daily $T_{max}$ values are reproduced relatively well by all models, which is also underlined by the rather low mean bias values in Table 1. Regarding the domain mean conditions of the overall characteristics, all models show reasonable performances for the heat wave frequency and mean duration, while this does not go for the mean intensity. The spatial agreement with the reference is not satisfactory for any RCM and section with the exception of the two WRF runs in the reproduction of the cumulative heat pattern for the 2003 event. In general, despite the same forcing by ERA-Interim, the RCMs exhibit a significant spread in their outputs. This is especially pronounced for the 2003 event, which underlines the difficulty of the models to reproduce single major events. Regarding the heat wave trends, the reference shows that, if there is a trend present, it is only increasing, indicating increases in the number of heat waves with time. The RCMs struggle with reproducing these trends. If trends are indicated, they are mostly not spatially accurate. All RCMs underestimate the proportion of grid cells with increasing trends. No specific physics scheme or configuration can be shown to be especially beneficial for the reproduction of the heat wave metrics. Furthermore, there seems to be no correlation between the RCM bias values (Table 1) and the respective RCM performances. According to the E-OBS reference, heat waves occurred about 31 times in the study period with an average duration of about 4 days and an average intensity of about 10 °C. This equals an average heat excess per day during a heat wave period of about 2.5 °C.

This analysis is a follow-up on our drought study (Petrovic et al., 2022). The two extreme event types, droughts and heat waves, are often regarded together and are indeed often, but not always related. We intended to investigate the same research questions for both events by employing and assessing the same model outputs by using the same or similar methods to work out commonalities, but above all differences between these two types of extreme events. In the drought study it was revealed that all RCMs performed on a similar level in reproducing the drought characteristics on domain average and that the WRF experiments showed clear benefits in the trend reproduction. In fact, only WRF was able to reproduce the observed trends in a fairly high spatial accuracy. This was mainly due to the model settings of WRF, but the higher resolution increased the spatial accuracy. In contrast to this, as shown in this study, there are more pronounced differences between the single RCM capabilities to reproduce heat wave characteristics, so that the choice of model is far more important here. Also in contrast to droughts,

there are no benefits of WRF in the trend reproduction. The two studies have in common that all RCMs were shown to struggle with the reproduction of the single major event of summer 2003. In both cases there is no model with a satisfying performance in this regard. Increased model resolution and tailored model settings were shown to be more important for drought than for heat wave simulation, especially for trends. This is most likely related to the different variables that play the crucial role for the respective type of extreme event: precipitation for droughts and (maximum) temperature for heat waves.

Our results suggest that a resolution of 12.5 km or even 15 km, as shown with the WRF@15 km run, is sufficient to reach similar findings to those obtained with finer resolutions. Furthermore, it is shown that the adjusted model settings to the specific target region of WRF only have limited impacts, suggesting that for the reproduction of $T_{max}$ and thus heat waves this is a less important factor. The results may guide in the selection of suitable RCMs for certain aspects of heat wave analysis in Germany and similar regions. Not only in a historical context, but also for future projections.

## Data Availability

The EURO-CORDEX data is freely available at the EURO-CORDEX website (https://www.euro-cordex.net/). The E-OBS data is freely available at the ECA&D website (https://www.ecad.eu/). The WRF data and the associated configuration files can be obtained online from Petrovic (2023, https://doi.org/10.5281/zenodo.7998809).

## Author contribution

DP, BF and HK developed the methodology for the study. DP carried out the data analysis and drafted the manuscript, with support of BF and HK. HK provided grant funding and supervised the research.

## Competing interests

The authors declare that they have no conflict of interest.

## Acknowledgements

The authors gratefully acknowledge the work of the WRF modeling community, the European Centre for Medium-Range Weather Forecast for the reanalysis data ERA-Interim, the contributors to the EURO-CORDEX projects used in this study, the ECA&D group for the E-OBS data set and Warscher et al. (2019) for providing the WRF simulation data. Big thanks go to Gerhard Smiatek for his support. This work is funded by the ClimXtreme project of the BMBF (German Federal Ministry of Education and Research) under grant "Förderkennzeichen 01LP1903J".

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
