# Peer review of "Heat wave characteristics: evaluation of regional climate model performances for Germany"

_Natural Hazards and Earth System Sciences, 2023_

## Author Comment (AC2)

**Response to RC2 on nhess-2023-91**

NOTE: Reviewer's comments are in black, our responses to the comments are given in blue below. Since we also address points in the body text here, we have highlighted the relevant text passages in bold for a better overview.

Petrovic et al. present an evaluation study of a set of reanalysis-driven regional climate model simulations with respect to the representation of heat wave characteristics over Germany. The authors employ simulated (EURO-CORDEX evaluation runs and two special WRF experiments) and observed (EOBS) daily maximum temperature in the period 1980-2099 and evaluate the model performance for several heat wave statistics (mean max. temperature, number of heat waves, heat wave duration, heat wave intensity). They find no clear benefit of the increase resolution in one of the WRF experiments and, in general, a mixed picture with no clear best or worse performing model. All models tend to underestimate the observed increase in the spatial extent of heat waves.

The manuscript is nicely written and contains a very good literature review and very appropriately outline the motivation for their study. The authors put great effort into documenting the model performances.

**However, I do not think that the manuscript is suited for publication in the journal as its novelty and its significance for a wider audience is very limited.**

As highlighted by both reviewers, heat waves (along with droughts) are a highly relevant topic for the general audience and also for the (high-resolution) modeling community. Both audiences are approached by this comprehensive evaluation. It provides a source of information for users of the model outputs, especially for those who work in a heat wave context. The manuscript reveals limitations of models to appropriately simulate past heat waves with associated characteristics. This also comes with implications for studies dealing with future projections as it shows that multi-model ensemble approaches are necessary. Furthermore, the study could guide in which models rather to use or not use for similar contexts, especially if only a limited number or set of models can be used. As for the novelty, we stress that a comparable study has not been performed yet for this region. Neither is there a study that addresses the two questions of model's resolution and settings influence on the heat wave reproduction that employs very high resolution (5 km) model outputs.

Generally, we would like to emphasize the fact that this study is a follow-up on the drought study published last year, also in NHESS within the same special issue. This common context needs to be considered, since droughts and heat waves are often regarded together. In the drought study key findings were among others that resolution and especially model settings do matter for certain aspects like trends and that all models performed on a similar level regarding the domain mean conditions for the characteristics. We wanted to investigate these questions also for heat waves by using the exact same data sets and applying the same or similar methods and draw conclusions about similarities and differences for these two types of extreme events. We were able to determine decisive differences especially regarding the role of resolution and model settings as well as the performance for the respective characteristics. For the heat waves there are considerable differences between the single model performances regarding the characteristics.

The paper has the flavour of a rather technical documentation of model performance and remains very descriptive. **Reasons for a good or bad reproduction of heat wave characteristics (large scale flow conditions, surface atmosphere interactions etc.) are discussed but actually not investigated.**

We agree that some investigation about potential reasons for the spread in the model performances would be useful. To address this, we will analyze additional variables like surface heat fluxes, soil moisture, radiation and sea level pressure and check the differences among the models and draw connections to the Tmax simulations. We will try to keep this short as we see the problem, that the manuscript may become too long. However, even if the manuscript is mainly descriptive, we still see a value in this. As mentioned above, this could guide in model selection for studies to be conducted in a similar context.

**The employed model ensemble is not special or up-to-date (though not yet outdated) and concerning the EURO-CORDEX sub-ensemble has been validated extensively in previous works.**

There was no claim that the employed model ensemble is some sort of special. The reason how this ensemble came about is that at the time of selection, back before the related drought study, only in the selected outputs were the required variables available for the required time period. After the drought study, we wanted to assess exactly the same model outputs regarding their ability to reproduce heat waves to work out commonalities, but above all differences. Regarding the validation of the ensemble in previous works, we are not aware that this was done in a heat wave context especially for the same study region. We want to stress that this goes beyond just validating the reproduction of a single variable like Tmean or Tmax, which may have been done in the literature. But analyzing heat wave reproduction is not only about the actual variable values but about the distributions and variability of values.

**Most recent experiments, for instance, employ the newer ERA5 reanalysis as boundary forcing.**

This is correct, as noted, for the most recent simulations. The time of model selection was a while ago. Back then, ERA-Interim driven outputs were much more frequent. In fact, even today there are mainly ERA-Interim driven reanalysis outputs available at the EURO-CORDEX platform. However, the decisive factor here is that the WRF experiments also obtained their boundary conditions by ERA-Interim. A comparison with EURO-CORDEX outputs appears to be only useful if the boundary forcing is the same for all model outputs, since this allows to draw conclusions about the core model performances.

**The resolution analysis is restricted to the WRF model and hence very limited, and the evaluation is carried out at the coarser 12 km resolution (i.e., the 5 km experiment cannot profit from its potentially finer-scale heat wave patterns in the evaluation).**

Correct, the resolution analysis is restricted to the WRF model. The very high resolution (5 km) of this experiment needs to be considered here. We did not have any other model outputs available in that high resolution. It also needs to be considered that usually in model grid resolution related studies the resolution of the EURO-CORDEX RCMs (12.5 km) is often among the highest in the literature. True, we chose to interpolate to the coarser grid, since it is common practice to regrid from the finer to the coarser grid. It would be unfair otherwise. That does not necessarily mean that all benefits of higher resolution are degraded. As shown in our drought paper, higher resolution benefits are still there. We rather conclude that the benefits of higher resolution in this context are not that pronounced.

**Moreover, most evaluated metrics actually neglect the mean bias (since evaluation is carried out with respect to the simulated 90th percentile or even as a normalized bias in the Taylor diagrams).**

The true mean bias values are given in Table 1 in the manuscript. Moreover, it is explicitly mentioned that "a connection between the Tmax bias (Table 1) and the overall performance does not seem to exist, since COSMO-CLM has the lowest mean bias value (-0.16 °C), but the worst overall

performance, while on the other hand RCA4, which holds the highest mean bias value (-2.40 °C), does not show a significant bad performance". It is also described that the respective 90th percentiles were calculated for each model output individually in order to "circumvent the Tmax biases among the different models".

That is why I believe that there is not much to learn from the paper. There are also number of minor issues listed below. My suggestion is hence to reject the submission for the time being, but to encourage the authors to produce a more up-to-date and process-based evaluation of heat wave characteristics instead. The topic is certainly highly relevant.

With kind regards

Naturally, we would consider this differently. Our study connects the findings of the drought analysis with this analysis of heat waves. While it has been astonishing that there is such a large spread between models to reproduce droughts, it is an important message that the regional models have considerable issues in resembling heat waves. We agree with the reviewer that the manuscript can be improved with respect to a more process-based analysis. However, switching to a completely new analysis of ERA-5 driven regional models would be out of scope of the current time-frame and funding constraints.

Minor issues:

- The issue of internal variability in different model domain sizes is not discussed at all (relating especially to the 2003 event, i.e. to a singular event). The domains of the WRF experiments are unclear and not shown.

We can see the point here and will add some discussion about the role of internal variability.

Correct, the domains of the WRF experiments are not shown. They can be found in the given reference Warscher et al. (2019), which is openly accessible under this link: https://www.mdpi.com/2073-4433/10/11/682.

- WRF@5km has been interpolated to the EUR-11 grid, i.e. the resolution has been degraded and any potential benefit in a better representation of finer-scale spatial heat wave patterns is not visible. Also, bilinear interpolation should be avoided when interpolating from a finer to a coarser grid (only a fraction of the high resolution grid cells is employed in this kind of interpolation). Better use a conservative remapping.

Regarding the interpolation from the 5 km to the 12.5 km grid, please see the comment above.

For similar cases where temperature data needs to be regridded from a finer to a coarser scale, bilinear interpolation appears to be common practice, e.g., in Vautard et al. (2013), Wang et al. (2019ab), Varela et al. (2020) and Machard et al. (2022) from the manuscript's bibliography. In contrast to precipitation, temperature is not a discontinuous variable, hence there is no mass that needs to be preserved. Oftentimes the interpolation method is not even mentioned, assuming that bilinear interpolation is even used more often. Further confirmation can for example be found on this NCAR info page: https://climatedataguide.ucar.edu/climate-tools/regridding-overview. Due to these reasons we would like to stick to the interpolation method currently applied.

- Taylor diagram: You have to consider that the RMSE shown by the green lines is the CENTERED RMSE, i.e. any mean bias is implicitly corrected for. This should at least be mentioned, or even shown

for instance by different sizes of the symbols (according to their mean bias). Also compare to Table 1 where the mean bias is shown.

Correct, it is the centered RMSE and this will be added to the text. We will also make connections to the values in Table 1.

- Figure 2: The shadings are hardly distinguishable from each other. It would be much better to show a simple curve for each model instead of a shaded curve.

We agree. We will work on the visibility of this figure.

- Chapter 4.2 and further: You often speak of a "cold" or "warm" bias here. I would strongly recommend to avoid this, as all statistics are related to the 90th percentile of each individual simulation. Any mean bias of the 90th percentile is implicitly subtracted and what you're looking at is actually the variability (temporal and in terms of magnitude) of all values above the simulated (or observed) 90th percentile.

We agree. We will replace "cold" and "warm" by "negative" and "positive", respectively.

- I suggest to speak of "WRF simulations" or "WRF experiments" instead of "WRF domains".

We agree. We will use the suggestions in the appropriate places in the text.

- Analysis of heat wave trends: You should definitely include a validation of Tmax (and its 90th percentile) in order to explain deficiencies in simulating heat wave trends.

We agree that this could be useful. We will add the grid cell based Tmax values averaged over the whole study period. Displaying the 90$^{th}$ percentile is considered difficult since it depends on the calendar day.

---

## Author Response (AR1)

**Author's response to the anonymous referee comments on nhess-2023-91**

Dear anonymous reviewers,

we sincerely thank you for the time and effort you took to review our manuscript and for providing constructive feedback and comments that will help improve the quality. The manuscript has been revised according to your suggestions. Please find below our response to each of your comments.

NOTE: Reviewer's comments are in black, our responses to the comments are given in blue below.

**Response to RC1 on nhess-2023-91**

The manuscript presents a study of the heat wave characteristics in Germany by using several RCM models and a WRF-based high-resolution downscaling dataset.

The topic is fascinating and fits the journal's aim and scope. Moreover, due to the last-year droughts, a general audience can be interested in such type of study. The manuscript is well-structured and organised. The English is fluent and understandable. Before considering the work for publication, I have the following points to raise with the authors:

1.     I understand the purpose of only comparing RCMs. Still, it might be interesting to exploit the newest version of ECMWF Reanalysis ERA5 as boundary conditions, increasing the spatial resolution to 0.25°. It means waiting until the new EURO-CORDEX simulation is performed, but at least WRF can be run in future studies comparing the previous and latest versions. In general, I would appreciate some discussion about that since one of the conclusions is that the RCM does not significantly improve performance.

We agree that the use of ERA5 as unified model forcing would possibly yield to more sophisticated models and that a comparison of an ERA5-driven WRF simulations with the current version would be interesting and may be conducted in future works.

We have added a passage on this topic to the discussion section.

2.     Is downscaling with WRF performed with two nested domains due to coarse ERA-Interim resolution (0.50° -> 15km -> 5km)? It should be explained in the data section; I didn't find the reference paper (Wagner and Kunstmann 2016).

Correct, two nests were used due to the coarse ERA-Interim resolution. We have added a sentence in the data section to make this clearer: "A two domain setup with one-way nesting was employed to downscale the ERA-Interim reanalysis of approx. 75 km."

The reference paper for the WRF simulations is actually Warscher et al. (2019). It is openly accessible under this link: https://www.mdpi.com/2073-4433/10/11/682.

3.     Figure 2. I need help seeing all the distribution, especially the reference E-OBS, which is completely missed. I suggest rethinking this.

We agree. We have worked on the visibility of this figure, especially making the E-OBS reference more visible.

4. Typos: L530 "are better"; L572 missing comma "(Petrovic et al., 2022), in which"; L611 "an average intensity"

We have corrected all the typos mentioned.

**Response to RC2 on nhess-2023-91**

Petrovic et al. present an evaluation study of a set of reanalysis-driven regional climate model simulations with respect to the representation of heat wave characteristics over Germany. The authors employ simulated (EURO-CORDEX evaluation runs and two special WRF experiments) and observed (EOBS) daily maximum temperature in the period 1980-2099 and evaluate the model performance for several heat wave statistics (mean max. temperature, number of heat waves, heat wave duration, heat wave intensity). They find no clear benefit of the increase resolution in one of the WRF experiments and, in general, a mixed picture with no clear best or worse performing model. All models tend to underestimate the observed increase in the spatial extent of heat waves.

The manuscript is nicely written and contains a very good literature review and very appropriately outline the motivation for their study. The authors put great effort into documenting the model performances.

**However, I do not think that the manuscript is suited for publication in the journal as its novelty and its significance for a wider audience is very limited.**

As highlighted by both reviewers, heat waves (along with droughts) are a highly relevant topic for the general audience and also for the (high-resolution) modeling community. Both audiences are approached by this comprehensive evaluation. It provides a source of information for users of the model outputs, especially for those who work in a heat wave context. The manuscript reveals limitations of models to appropriately simulate past heat waves with associated characteristics. This also comes with implications for studies dealing with future projections as it shows that multi-model ensemble approaches are necessary. Furthermore, the study could guide in which models rather to use or not use for similar contexts, especially if only a limited number or set of models can be used. As for the novelty, we stress that a comparable study has not been performed yet for this region. Neither is there a study that addresses the two questions of model's resolution and settings influence on the heat wave reproduction that employs very high resolution (5 km) model outputs.

Generally, we would like to emphasize the fact that this study is a follow-up on the drought study published last year, also in NHESS within the same special issue. This common context needs to be considered, since droughts and heat waves are often regarded together. In the drought study key findings were among others that resolution and especially model settings do matter for certain aspects like trends and that all models performed on a similar level regarding the domain mean conditions for the characteristics. We wanted to investigate these questions also for heat waves by using the exact same data sets and applying the same or similar methods and draw conclusions about similarities and differences for these two types of extreme events. We were able to determine decisive differences especially regarding the role of resolution and model settings as well as the performance for the respective characteristics. For the heat waves there are considerable differences between the single model performances regarding the characteristics.

Since also mentioned and requested by the editor, we have added a part to the conclusions section to make the connection between the two studies more clear.

The paper has the flavour of a rather technical documentation of model performance and remains very descriptive. **Reasons for a good or bad reproduction of heat wave characteristics (large scale flow conditions, surface atmosphere interactions etc.) are discussed but actually not investigated.**

We agree that some investigation about potential reasons for the spread in the model performances would be useful. To address this, we have analyzed the following additional variables from the individual EURO-CORDEX outputs: sensible and latent heat flux, incoming and outgoing short and

long wave radiation, soil moisture and surface pressure. We could only use the outputs from the EURO-CORDEX RCMs, other outputs from the WRF experiments were no longer available. We tried to identify Tmax dependencies on the other variables by performing correlation analysis (Figs. S1 and S3). Afterwards we compared the distributions of the respective variables among the models in order to find significant differences at those variables, that showed a high correlation with Tmax, by examining boxplots (Figs. S2 and S4). We did this for the summer months of the entire study period, and also only for the summer period of 2003 to ensure that heat wave conditions are included in the analysis. In both cases there is no clear evidence of potential reasons behind the bias in the heat wave reproduction, since the distributions of the variables that showed highest Tmax correlations are quite similar among the models. The only exception is soil moisture and to some degree also the sensible heat flux, which did not show high Tmax correlations. The comparison of soil moisture among different models does not seem useful, since there are considerable differences in the modeled soils among the models, e.g., different number of soil layers with different layer depths. Anyways, soil moisture does not have high correlations with Tmax. Since the time course of the variables is decisive and the consideration of the distributions allows only little statements about this, we have additionally looked at the spatially averaged courses of the individual variables for each model run for the summer 2003 period (Fig. S5). Largest differences between the individual models are found for the soil moisture here, too, which is not very meaningful, as described above. Naturally, there is a spread between the single lines for each variable, but they greatly agree in their patterns or variability, respectively, which is more important than the agreement of values. It is not possible to establish a consistency with the actual study results, as, for example, the model that shows highest overestimation in heat wave characteristics, COSMO-CLM, does not stand out in this view. Thus, the differences that occur in part, especially for the radiation variables since they show highest correlations, may provide explanation only to some degree. The resulting figures were added in a supplement, as they only provide supporting information. Other potential explanations for the biases like different land use data and also internal variability is beyond our possibilities to investigate. As described by other authors, the biases remain difficult to explain and to find reasons for. We have added a passage on that to the discussion section.

**The employed model ensemble is not special or up-to-date (though not yet outdated) and concerning the EURO-CORDEX sub-ensemble has been validated extensively in previous works.**

There was no claim that the employed model ensemble is some sort of special. The reason how this ensemble came about is that at the time of selection, back before the related drought study, only in the selected outputs were the required variables available for the required time period. After the drought study, we wanted to assess exactly the same model outputs regarding their ability to reproduce heat waves to work out commonalities, but above all differences. Regarding the validation of the ensemble in previous works, we are not aware that this was done in a heat wave context especially for the same study region. We want to stress that this study goes beyond just validating the reproduction of single variables like Tmean or Tmax, which may have been available in the literature. Studying the reproduction of heat waves is more than just analyzing the reproduction of the actual variable values. The temporal course, variability, and distributions of the values play a critical role in this context. Analysis and inclusion of these factors may draw a very different picture about the model capabilities than a mere analysis of individual variables would allow.

**Most recent experiments, for instance, employ the newer ERA5 reanalysis as boundary forcing.**

This is correct, as noted, for the most recent simulations. The time of model selection was a while ago. Back then, ERA-Interim driven outputs were much more frequent. In fact, even today there are only ERA-Interim driven reanalysis outputs available at the EURO-CORDEX platform. However, the decisive factor here is that the WRF experiments also obtained their boundary conditions by ERA-

Interim. A comparison with EURO-CORDEX outputs appears to be only useful if the boundary forcing is the same for all model outputs, since only such a study setup allows to draw conclusions about the core model performances.

**The resolution analysis is restricted to the WRF model and hence very limited, and the evaluation is carried out at the coarser 12 km resolution (i.e., the 5 km experiment cannot profit from its potentially finer-scale heat wave patterns in the evaluation).**

Correct, the resolution analysis is restricted to the WRF model. The very high resolution (5 km) of this experiment needs to be considered here. We did not have any other model outputs available in that high resolution. It also needs to be considered that usually in model grid resolution related studies the resolution of the EURO-CORDEX RCMs (12.5 km) is often among the highest in the literature. True, we chose to interpolate to the coarser grid, since it is common practice to regrid from the finer to the coarser grid. It would be unfair otherwise. That does not necessarily mean that all benefits of higher resolution are degraded. As shown in our drought paper, higher resolution benefits are still there. We rather conclude that the benefits of higher resolution in this context are not that pronounced.

**Moreover, most evaluated metrics actually neglect the mean bias (since evaluation is carried out with respect to the simulated 90th percentile or even as a normalized bias in the Taylor diagrams).**

The true mean bias values are given in Table 1 in the manuscript. Moreover, it is explicitly mentioned that "a connection between the Tmax bias (Table 1) and the overall performance does not seem to exist, since COSMO-CLM has the lowest mean bias value (-0.16 °C), but the worst overall performance, while on the other hand RCA4, which holds the highest mean bias value (-2.40 °C), does not show a significant bad performance". It is also described that the respective 90[th] percentiles were calculated for each model output individually in order to "circumvent the Tmax biases among the different models".

That is why I believe that there is not much to learn from the paper. There are also number of minor issues listed below. My suggestion is hence to reject the submission for the time being, but to encourage the authors to produce a more up-to-date and process-based evaluation of heat wave characteristics instead. The topic is certainly highly relevant.

With kind regards

Naturally, we would consider this differently. Our study connects the findings of the drought analysis with this analysis of heat waves. While it has been astonishing that there is such a large spread between models to reproduce droughts, it is an important message that the regional models have considerable issues in resembling heat waves. We agree with the reviewer that the manuscript can be improved with respect to a more process-based analysis. However, switching to a completely new analysis of ERA-5 driven regional models would be out of scope of the current time-frame and funding constraints. Moreover, as mentioned above, there is currently no extensive set of ERA-5 driven simulations available.

Minor issues:

- The issue of internal variability in different model domain sizes is not discussed at all (relating especially to the 2003 event, i.e. to a singular event). The domains of the WRF experiments are unclear and not shown.

We can see the point here and have added a passage on the role of internal variability to the discussion section.

Correct, the domains of the WRF experiments are not shown. They can be found in the given reference Warscher et al. (2019), which is openly accessible under this link: https://www.mdpi.com/2073-4433/10/11/682.

- WRF@5km has been interpolated to the EUR-11 grid, i.e. the resolution has been degraded and any potential benefit in a better representation of finer-scale spatial heat wave patterns is not visible. Also, bilinear interpolation should be avoided when interpolating from a finer to a coarser grid (only a fraction of the high resolution grid cells is employed in this kind of interpolation). Better use a conservative remapping.

Regarding the interpolation from the 5 km to the 12.5 km grid, please see the comment above.

For similar cases where temperature data needs to be regridded from a finer to a coarser scale, bilinear interpolation appears to be common practice, e.g., in Vautard et al. (2013), Wang et al. (2019ab), Varela et al. (2020) and Machard et al. (2022) from the manuscript's bibliography. In contrast to precipitation, temperature is not a discontinuous variable, hence there is no mass that needs to be preserved. Oftentimes the interpolation method is not even mentioned, assuming that bilinear interpolation is even used more often. Further confirmation can for example be found on this NCAR info page: https://climatedataguide.ucar.edu/climate-tools/regridding-overview. Based on these reasons we would like to stick to the interpolation method currently applied.

- Taylor diagram: You have to consider that the RMSE shown by the green lines is the CENTERED RMSE, i.e. any mean bias is implicitly corrected for. This should at least be mentioned, or even shown for instance by different sizes of the symbols (according to their mean bias). Also compare to Table 1 where the mean bias is shown.

Correct, it is the centered RMSE and it has been added to the text. We have also made connections to the values in Table 1.

- Figure 2: The shadings are hardly distinguishable from each other. It would be much better to show a simple curve for each model instead of a shaded curve.

We agree. We have adjusted the figure according to the suggestion.

- Chapter 4.2 and further: You often speak of a "cold" or "warm" bias here. I would strongly recommend to avoid this, as all statistics are related to the 90th percentile of each individual simulation. Any mean bias of the 90th percentile is implicitly subtracted and what you're looking at is actually the variability (temporal and in terms of magnitude) of all values above the simulated (or observed) 90th percentile.

We agree. We have replaced "cold" and "warm" by "negative" and "positive", respectively.

- I suggest to speak of "WRF simulations" or "WRF experiments" instead of "WRF domains".

We agree. We have used the suggestions as well as "outputs" in the appropriate places in the text.

- Analysis of heat wave trends: You should definitely include a validation of Tmax (and its 90th percentile) in order to explain deficiencies in simulating heat wave trends.

We have analyzed the Tmax and Tmax 90$^{th}$ percentile patterns averaged for the summer months over the whole study period. As it turned out, no conclusions could be drawn about trends or the capability to reproduce trends. Generally, we do not really see the point here. The section is about

the trend in the number of annual heat waves. Proper trend reproduction does not necessarily imply the reproduction of accurate Tmax values. It is about the consecutive number of hot days. This refers, as described in the manuscript, to the values from the respective simulation, not to the values from the observations. Thus, for proper trend reproduction it is necessary to simulate hotter than "normal" conditions. Both, "hotter" and "normal", differ among the individual models and observations. Nevertheless, they can show the same trend.

---

## Author Response (AR3)

**Author's response to the anonymous referee comments on nhess-2023-91**

Dear anonymous reviewer,

we sincerely thank you for the time and effort you took to review our manuscript and for providing constructive feedback and comments that will help improve the quality. The manuscript has been revised according to your suggestions. Please find below our response to each of your comments.

NOTE: Reviewer's comments are in black, our responses to the comments are given in blue below.

**Response to RC3 on nhess-2023-91**

After reading the manuscript, I found the experimental setup and results convincing but, in my opinion, the discussion section is hard to follow and needs to be more concise. In the revised manuscript, a lot on new text has been added into this section probably based on reviewers' suggestions (I have not participated on the first round of revisions). In my opinion, the authors focus too much on explaining individual models' biases in heat waves' simulation and these explanations are often not very conclusive and sometimes even contradicting each other. Unfortunately, it is difficult to provide specific comments how to improve this section but the length of discussion (nearly 3,500 words not counting conclusions) is in my opinion not justifiable. Maybe focusing more on WRF@5km (main message of the manuscript) could provide a more concise discussion.

We can see the point that the discussion appears to be too comprehensive. We have shortened this section and moved part of it to the results section (see below). As a result, the section is now less than 2,500 words long.

Other comments:
Line 22: 'Maximum temperature was reproduced reasonably well by all models' – then why calculating the 90th percentiles individually for each data set to circumvent the biases (Line 144)? It also contradicts the sentence in Line 236. I suggest revisiting this sentence.

We agree and can see the reason for confusion. This is why we adjusted the sentence in Line 22 to: "Maximum temperature was only reproduced satisfactorily by some models".

Line 49: Lhotka et al. (2017) → Lhotka et al. (2018a)

Correct, we have adjusted this.

Line 215: 'RACMO is the best performing EURO-CORDEX RCM, ALADIN the worst.' please add in what metric for better clarity.

We have adjusted the sentence to "RACMO is the best performing EURO-CORDEX RCM **in all of the three categories (correlation, CRMSE and standard deviation match)**, ALADIN the worst" for more clarity.

Table 1: Authors may consider adding Tmax bias values for 90th, 95th, and 99th percentiles, which may help interpreting the inter-model differences in heat waves' simulations.

We do not consider this as very useful, since these percentile bias values would refer to the overall time series, while in this context it is about the percentile values for the respective days based on the whole time period.

Line 528: 'an overestimation of which metrics? I suppose it should be 'persistence' (?) Please clarify.

Correct, we have added this for more clarification.

Lines 530–552: In my opinion, this newly added information belong to the results section.

We agree on this. Since this information refers to the bias in general, we have added an extra section in the results for that. This also shortens the discussion.

Line 615: 'overall best performances' should be defined, otherwise it is a subjective metric.

We agree. We have changed the sentence to "The three RCMs with the overall best performances **especially regarding the reproduction of heat wave characteristics** are ALADIN, REMO and WRF@15 km".

Line 680: Authors should mention both 5 an 15km WRF runs.

We agree. We have adjusted the sentence to: "For this purpose, we employ an ensemble of six ERA-Interim-driven EURO-CORDEX RCMs of 12.5 km horizontal grid resolution **as well as outputs of a target area tailored, ERA-Interim-driven WRF simulation at 5 and 15 km resolution**".